# Analysis of pulsed cisplatin signalling dynamics identifies effectors of resistance in lung adenocarcinoma

Jordan F Hastings[1†], Alvaro Gonzalez Rajal[1†], Sharissa L Latham[1,2], Jeremy ZR Han[1], Rachael A McCloy[1], Yolande EI O'Donnell[1], Monica Phimmachanh[1], Alexander D Murphy[3], Adnan Nagrial[3], Dariush Daneshvar[3], Venessa Chin[1,2,4], D Neil Watkins[5,6,7,8], Andrew Burgess[9,10]*, David R Croucher[1,2,11]*

[1]The Kinghorn Cancer Centre, Garvan Institute of Medical Research, Sydney, Australia; [2]St Vincent's Hospital Clinical School, University of New South Wales, Sydney, Australia; [3]Crown Princess Mary Cancer Centre, Westmead and Blacktown Hospitals, Sydney, Australia; [4]St Vincent's Hospital Sydney, Darlinghurst, Australia; [5]Hudson Institute of Medical Research, Victoria, Australia; [6]Department of Molecular and Translational Medicine, School of Medicine, Nursing and Health Sciences, Monash University, Victoria, Australia; [7]Research Institute in Oncology and Hematology, Cancer Care Manitoba, Winnipeg, Canada; [8]Department of Internal Medicine, Rady Faculty of Health Science, University of Manitoba, Winnipeg, Canada; [9]ANZAC Research Institute, Concord, Australia; [10]The University of Sydney Concord Clinical School, Faculty of Medicine and Health, Sydney, Australia; [11]School of Medicine, University College Dublin, Belfield, Dublin, Ireland

*For correspondence:
andrew.burgess@sydney.edu.au
(AB);
d.croucher@garvan.org.au (DRC)

†These authors contributed equally to this work

Competing interests: The authors declare that no competing interests exist.

**Abstract** The identification of clinically viable strategies for overcoming resistance to platinum chemotherapy in lung adenocarcinoma has previously been hampered by inappropriately tailored in vitro assays of drug response. Therefore, using a pulse model that closely mimics the in vivo pharmacokinetics of platinum therapy, we profiled cisplatin-induced signalling, DNA-damage and apoptotic responses across a panel of human lung adenocarcinoma cell lines. By coupling this data to real-time, single-cell imaging of cell cycle and apoptosis we provide a fine-grained stratification of response, where a P70S6K-mediated signalling axis promotes resistance on a *TP53* wildtype or null background, but not a mutant *TP53* background. This finding highlights the value of in vitro models that match the physiological pharmacokinetics of drug exposure. Furthermore, it also demonstrates the importance of a mechanistic understanding of the interplay between somatic mutations and the signalling networks that govern drug response for the implementation of any consistently effective, patient-specific therapy.

## Introduction

Lung adenocarcinoma is the most common form of lung cancer, the leading cause of cancer-related death worldwide. Lung adenocarcinoma is typically diagnosed late, meaning that most patients require systemic chemotherapy (*Chen et al., 2014*). Platinum-based chemotherapy is likely to remain an important treatment modality for these patients due to the emergence of resistance to targeted therapies in EGFR, ALK or ROS mutant tumours (*Lindeman et al., 2018*), and the fact that most patients do not respond to single agent immunotherapy (*Kim et al., 2019*; *Doroshow et al., 2019*).

**eLife digest** Lung adenocarcinoma is the most common type of lung cancer, and it emerges because of a variety of harmful genetic changes, or mutations. Two lung cancer patients – or indeed, two different sets of cancerous cells within a patient – may therefore carry different damaging mutations.

A group of drugs called platinum-based chemotherapies are currently the most effective way to treat lung adenocarcinoma. Yet, only 30% of patients actually respond to the therapy. Many studies conducted in laboratory settings have tried to understand why most cases are resistant to treatment, with limited success.

Here, Hastings, Gonzalez-Rajal et al. propose that previous research has been inconclusive because studies done in the laboratory do not reflect how the treatment is actually administered. In patients, platinum-based drugs are cleared from the body within a few hours, but during experiments, the treatment is continually administered to cells growing in a dish.

Hastings, Gonzalez-Rajal et al. therefore developed a laboratory method that mimics the way cells are exposed to platinum-based chemotherapy in the body. These experiments showed that the lung adenocarcinoma cells which resisted treatment also carried high levels of a protein known as P70S6K. Pairing platinum-based chemotherapy with a drug that blocks the activity of P70S6K killed these resistant cells. This combination also treated human lung adenocarcinoma tumours growing under the skin of mice. However, it was ineffective on cancerous cells that carry a mutation in a protein called p53, which is often defective in cancers.

Overall, this work demonstrates the need to refine how drugs are tested in the laboratory to better reflect real-life conditions. It also underlines the importance of personalizing drug combinations to the genetic background of each tumour, a concept that will be vital to consider in future clinical trials.

Despite the use of platinum-based chemotherapy in lung adenocarcinoma for over four decades, response rates remain below 30% due to the prevalence of innate resistance (*Pilkington et al., 2015*; *Bonanno et al., 2014*). In addition, dose-related nephrotoxicity remains a challenge in many patients (*Pabla and Dong, 2008*). Strategies to improve platinum efficacy could therefore significantly improve outcomes for lung adenocarcinoma patients. However, unravelling platinum resistance in lung adenocarcinoma has proven challenging, as over 147 mechanisms of resistance have been proposed (*Stewart, 2007*), yet there remains a lack of viable clinical options to improve response rates.

From an experimental viewpoint, discordance between the in vivo pharmacokinetics of platinum chemotherapies and their use within in vitro assays has likely contributed to the identification of putative resistance mechanisms and drug targets that have not ultimately translated to the clinic. Traditionally, in vitro methods for the investigation of drug response have involved culturing cancer cells in the continuous presence of high-dose chemotherapy over several days. This contrasts with pharmacokinetic studies in humans and rodents demonstrating that both cisplatin and carboplatin are rapidly cleared from the circulation, and the tumour, within 2–3 hr following administration (*Andersson et al., 1996*; *Johansen et al., 2002*). Therefore, in this study we have utilised a cisplatin pulse model, which more closely recapitulates these physiological pharmacokinetics, aiming to maintain the fidelity of the apoptotic mechanism mediated by cisplatin in vivo.

We have previously shown that predictive computational models of drug-induced apoptotic signalling dynamics can be used as a prognostic indicator of neuroblastoma patient survival (*Fey et al., 2015*). To move towards a similar concept to platinum resistance in lung adenocarcinoma, we now present an in-depth analysis of the dynamic signalling response to a pulse of platinum chemotherapy, describing the relationship between a number of key signalling nodes, the DNA damage response and platinum sensitivity. Importantly, we also propose a therapeutic strategy targeting P70S6K using the dual PI3K/mTOR inhibitor dactolisib, with the potential to improve the efficacy of current platinum-based treatment regimens.

## Results

### Continuous versus pulsed cisplatin treatment

In order to directly compare the response of lung adenocarcinoma cells to the continuous presence of cisplatin, or a pulse of cisplatin that mimics in vivo pharmacokinetics (2 hr, 5 μg/mL) (*Figure 1—figure supplement 1A*), we monitored the growth and apoptosis of the innately resistant A549 lung adenocarcinoma cell line (*Marini et al., 2018*) by both live cell imaging (*Figure 1—figure supplement 1B*) and a cell viability assay (*Figure 1—figure supplement 1C*), under both conditions. This analysis demonstrated that while continuous exposure to cisplatin resulted in decreased cell number and increased apoptosis over 72 hr, a pulse of cisplatin only reduced the rate of cell proliferation and did not induce apoptosis.

To further examine the differences between these two models, we used multiplexed, bead-based protein analysis to investigate the DNA damage, apoptotic and signalling response for key pathway components previously implicated in the response to continuous cisplatin exposure (*Stewart, 2007*; *Marini et al., 2018*; *Supplementary file 1*, *Figure 1—figure supplement 1D*). As might be expected, the continuous exposure model resulted in a significantly elevated and sustained DNA damage response when compared to the pulse model, particularly for the phosphorylation of Chk2 (Ser345), p53 (Ser15 and Ser46), pH2A.X (Ser139 - γH2A.X) and expression of p21 and MDM2 (*Figure 1—figure supplement 2*). This heightened DNA damage response during the continuous exposure to cisplatin was also reflected in the increased activation of Caspase 3, which was completely absent for the pulse model. Furthermore, while p38 and ERK activation were significantly increased in cells continuously exposed to cisplatin, the expression of MCL-1 and detection of MCL-1/Bak dimers only significantly increased in cells treated with a pulse of cisplatin. This finding demonstrates that not only does the continuous exposure model result in a DNA damage and apoptotic response that is incongruent with that observed following a pulse of cisplatin, the dynamics of key signalling pathways are fundamentally different between these two treatment models. Taken together, this data demonstrates that previous mechanisms of platinum resistance established using a continuous exposure model should be reconsidered in the light of these new findings regarding the response to physiological levels of drug exposure.

### Response to cisplatin is not associated with either TP53 status or drug-efflux

To investigate potential mechanisms of platinum resistance using a model consistent with the physiological pharmacokinetics of platinum therapy (*Figure 1A*), we applied this cisplatin pulse model to a panel of six lung adenocarcinoma cell lines with distinct *TP53* mutation backgrounds (two wildtype lines, two mutant *TP53* lines and two *TP53* null) and measured the apoptotic response at 72 hr (*Figure 1B*). Based upon this model we observed a range of sensitivity to cisplatin, from the most resistant A549 line (~3% apoptosis) to the most responsive NCI-H1299 line (~32% apoptosis). However, these cell lines could not be stratified simply according to their *TP53* mutation status, or other frequently observed genetic alterations (*Supplementary file 2*).

As the action of drug-efflux pumps is another commonly proposed mechanism of resistance to platinum therapy (*Hoffmann and Lambert, 2014*), we performed fluorescence microscopy with an antibody towards cisplatin-induced DNA adducts at multiple time-points following a 2 hr cisplatin pulse (*Figure 1C*). This analysis demonstrated that within this model, all six cell lines displayed significant nuclear localised cisplatin-DNA adducts following a 2 hr pulse (*Figure 1C*, *Figure 1—figure supplement 3*), suggesting that drug efflux is not associated with variations in the apoptotic response to a pulse of cisplatin in these lines. Furthermore, these cisplatin-DNA adducts progressively resolved over a 72 hr period in all cell lines (*Figure 1C*), confirming that pathways responsible for facilitating the removal of cisplatin adducts are also functional across this panel.

### Multi-dimensional analysis of cisplatin-induced signalling dynamics

To gain an understanding of the wider signalling networks associated with resistance to platinum therapy, we utilised multiplexed, magnetic bead based assays to profile the signalling and DNA damage response at multiple time points following a 2 hr pulse of cisplatin, across all six cell lines (*Figure 1D*). For this analysis, we tracked the dynamics of 47 different protein analytes

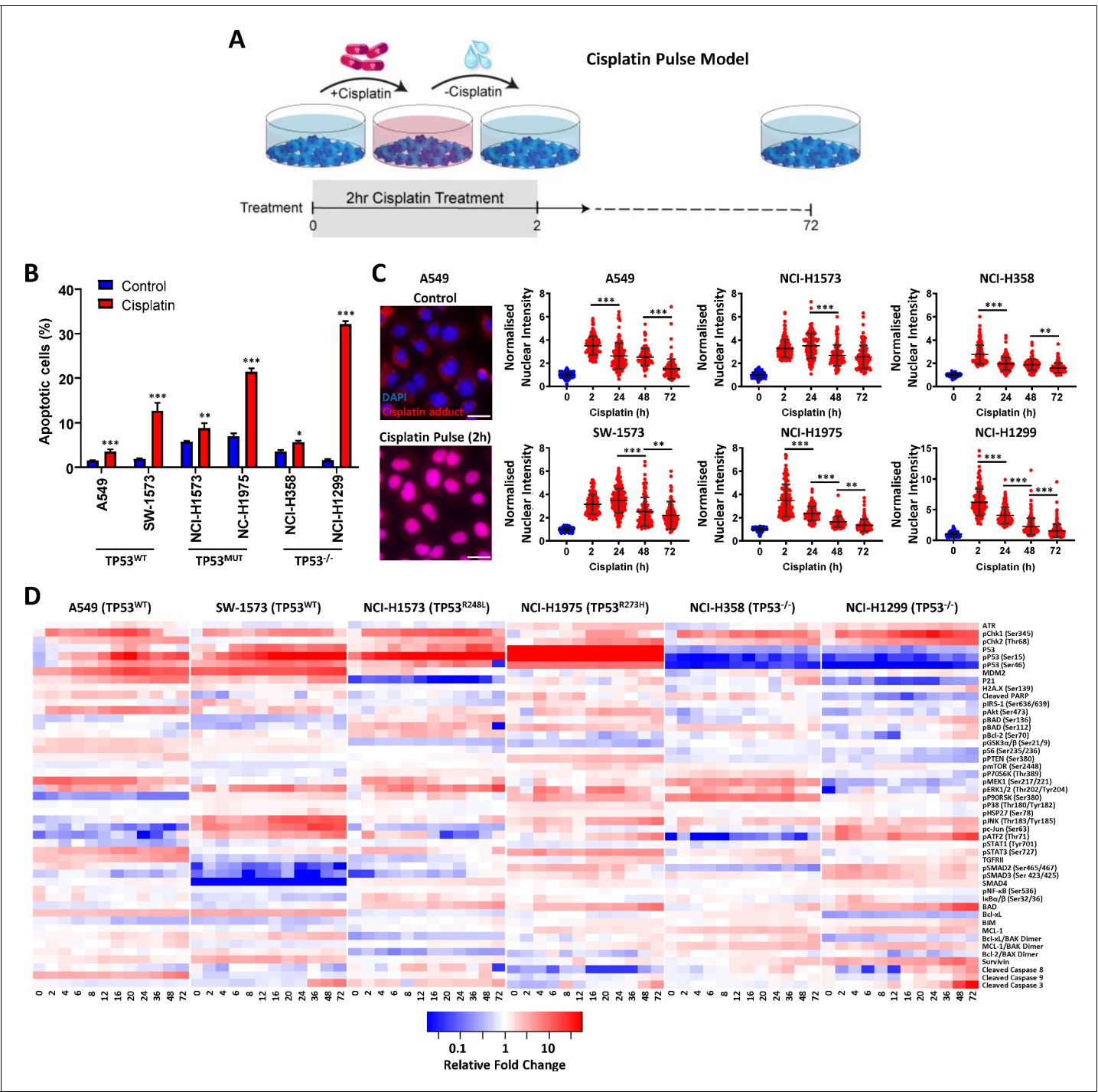

**Figure 1.** Multiplexed analysis of cisplatin-induced signalling. (**A**) Schematic of the cisplatin pulse model (5 μg/mL, 2 hr) and continuous pulse model (5 μg/mL, 72 hr). (**B**) Apoptosis measured by propidium iodide staining for the sub-G1 population, performed 72 hr following a cisplatin pulse across a panel of lung adenocarcinoma cell lines, as indicated (n = 3, mean ± SD). Statistical significance was determined by t-test (***p<0.001, **p<0.01, *p<0.05). (**C**) Representative images of anti-cisplatin antibody staining in A549 cells following a cisplatin pulse, and quantification of nuclear cisplatin-DNA adducts across the cell line panel (n ≥ 100, mean ± SD). Nuclear staining intensity was normalized to background, cytoplasmic staining within each cell line. Statistical significance was determined by one-way ANOVA (***p<0.001, **p<0.01). All treatment conditions (red) are significantly different from control (blue), p<0.001. (**D**) Multiplexed analysis of DNA damage, apoptosis and signalling pathways following a cisplatin pulse across a panel of lung adenocarcinoma cell lines, as indicated (n = 3, mean).

The online version of this article includes the following source data and figure supplement(s) for figure 1:

**Source data 1.** Summary of the analytes used for multiplex signalling analysis.

*Figure 1 continued on next page*

**Figure supplement 1.** Continuous versus pulsed cisplatin treatment of A549 cells.
**Figure supplement 2.** Continuous versus pulsed cisplatin treatment of A549 cells.
**Figure supplement 3.** Imaging of cisplatin-DNA adducts.
**Figure supplement 4.** p53 pathway dynamics.

(*Supplementary file 3*) over a 72 hr period, focusing on elements of signalling network structures that we recently implicated in platinum chemoresistance (*Marini et al., 2018*), including the MAPK, PI3K/mTOR, NF-κB and TGFβ pathways, as well as a number of key apoptotic mediators and DNA damage response proteins (*Figure 1D*).

From this dataset, a clear correlation can be seen between the *TP53* mutation status of each cell line and the dynamics of the p53 pathway (*Figure 1—figure supplement 4*), which validates the fidelity of the multiplexed platform for this type of analysis. In line with the detection of cisplatin-DNA adducts (*Figure 1C*), Chk1 (Ser345) phosphorylation increased rapidly in all lines following the 2 hr pulse, followed by a slower wave of Chk2 (Thr68) phosphorylation in all lines except SW-1573. Within both the *TP53* wildtype cell lines (A549 and SW-1573) this was followed by phosphorylation of p53 (Ser15 and Ser46), the accumulation of total p53 and increased expression of the p53 transcriptional targets MDM2 and p21. In the *TP53* mutant cell lines (NCI-H1573 and NCI-H1975) p53 phosphorylation still occurred, however in line with their loss of DNA binding capability, this did not result in increased expression of MDM2 or p21. As would be expected for the *TP53* null lines (NCI-H358 and NCI-H1299), p53 is absent and therefore not detected in either the total or phosphorylated form. Interestingly though, there was a significant increase in p21 and MDM2 expression in the NCI-H358 line in the absence of p53 expression.

As we had already determined that *TP53* status alone was not sufficient to explain resistance to cisplatin (*Figure 1B*), we further analysed the whole dataset by performing a principal component analysis (PCA) (*Figure 2*). This form of dimensionality reduction can be used to identify correlative relationships between variables within a large dataset, and here we have used it to create a visual representation of the association between key signalling nodes and the response to cisplatin across the entire cell line panel. Using this multi-dimensional analysis, we were able to capture ~70% of variance in the dataset within the first four principal components (*Figure 2—figure supplement 1A*). Unsurprisingly, plotting the first two principal components (PC1 and PC2) against each other (*Figure 2A*) resulted in separation of the cell lines primarily according to their *TP53* status. As might be expected, within PC1 and PC2 the *TP53* wild-type A549 and SW-1573 lines associated with higher p21 and MDM2 expression, while the *TP53* mutant lines separated from the *TP53* null lines mostly on the basis of higher p53 expression and phosphorylation levels (*Figure 2A,B,C*). However, plotting the third and fourth principal components (PC3 and PC4, *Figure 2D,E*) created a clear delineation between the three most resistant cell lines (A549, NCI-H358 and NCI-H1573), which cluster towards the left hand side of PC3 (x-axis), and the three most sensitive lines (SW-1573, NCI-1975 and NCI-H1299) which move progressively along PC3 over the 72 hr timeframe.

## Validation of model-based observations

As platinum chemotherapies work by forming covalent DNA adducts, which distort the DNA helix and block replication, the progressive accumulation of single stranded and double stranded breaks is thought to induce apoptosis (*Jamieson and Lippard, 1999*). This is in line with the movement of cisplatin sensitive cell lines along PC3 (x-axis) towards higher levels of γH2A.X (H2A.X$^{S139}$), cleaved caspase 3 and a stress associated MAPK signalling axis (pATF2, pJNK, pc-Jun) (*Figure 2D,E*). The association between this signalling state and increased sensitivity to cisplatin can also be clearly observed by overlaying an orthogonal readout of the apoptotic response onto this PCA plot (*Figure 3A*). This real-time apoptosis data was generated using live-cell imaging with a fluorescent caspase substrate as an indicator of cell death across the cell line panel for 72 hr following the cisplatin pulse treatment.

Using this approach, we now created a visual representation that both reflects the variance within the original dataset and demonstrates the key signalling nodes that are associated with differing degrees of platinum-induced apoptosis. While increasing levels of apoptosis are observed over time

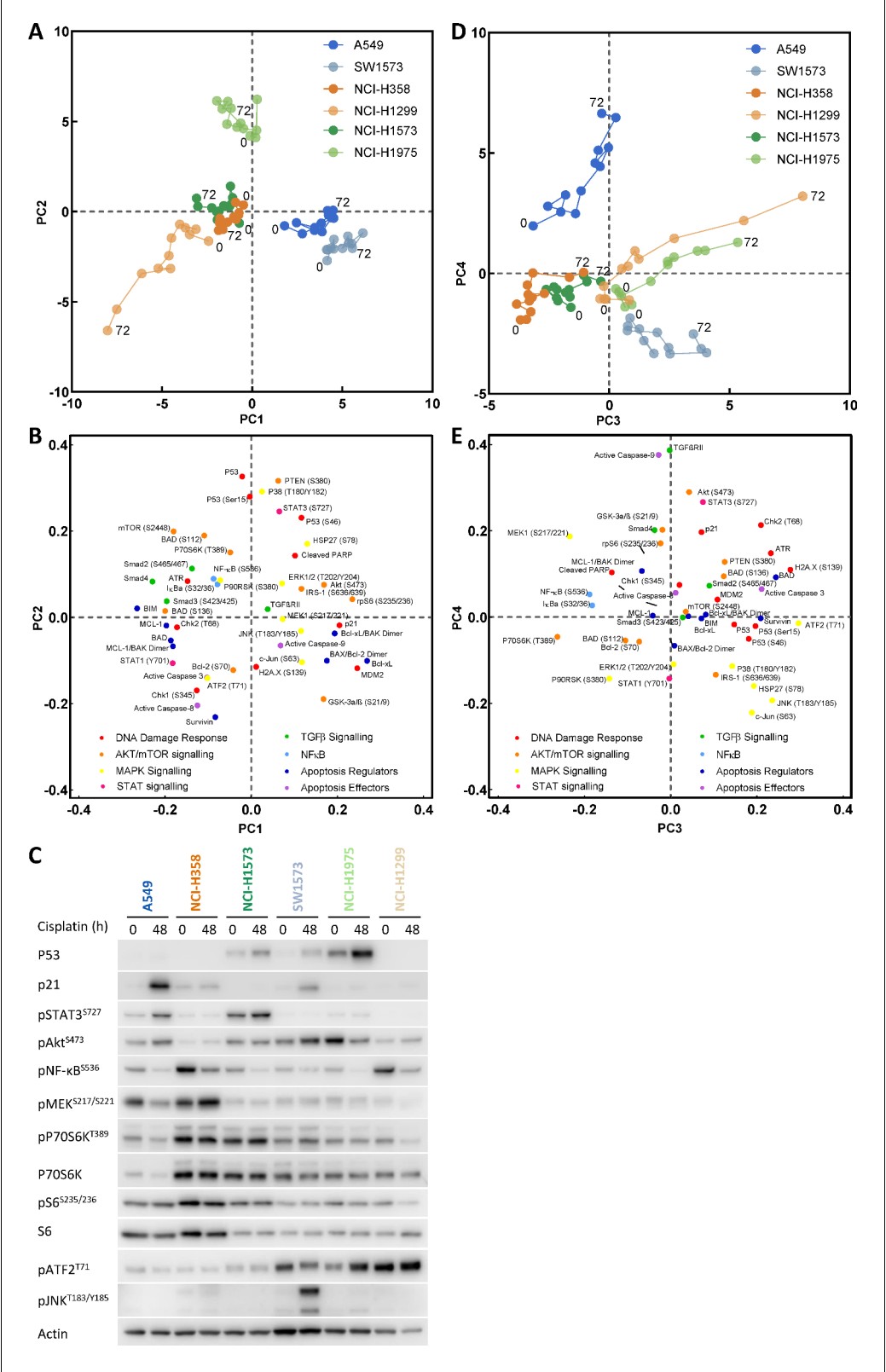

**Figure 2.** Principal component analysis. (**A**) Visualisation of principal component 1 (PC1) against component 2 (PC2) for the principal component analysis of cisplatin induced signalling across the cell line panel. (**B**) Distribution of the analytes according to their weighting within PC1 and PC2. (**C**) Western blotting for selected analytes across the cell panel prior to, and 48 hr post a cisplatin pulse. (**D**) Visualisation of principal component 3 (PC3)

*Figure 2 continued on next page*

*Figure 2 continued*

against component 4 (PC4) for the principal component analysis of cisplatin induced signalling across the cell line panel. (**E**) Distribution of the analytes according to their weighting within PC3 and PC4.

The online version of this article includes the following figure supplement(s) for figure 2:

**Figure supplement 1.** Extended analysis of the PCA data.

in the three sensitive cell lines (SW-1573, NCI-1975 and NCI-H1299), the three resistant cell lines display significantly lower levels of apoptosis (*Figure 2—figure supplement 1B*) and do not move along PC3 towards the region of DNA damage and apoptosis. Instead the NCI-H358 and NCI-1573 lines remain towards the left hand side of PC3, in a region characterised by higher phosphorylation of the mTOR pathway component P70S6K (Thr389), the MAPK components MEK1 (Ser217/221) and P90RSK (S380), NF-κB (S536) and IκBα (Ser32/36) (*Figure 2D,E*). The resistant A549 line also remains shifted towards the left of PC3, although also moves up PC4 towards a region with higher expression of TGFβRII, cleaved caspase 9, pAkt (Ser473) and pSTAT3 (Ser727) (*Figure 2D,E*). The association of elevated pMEK (Ser217/221), pNF-κB (S536), pSTAT3 (Ser727) and pP70S6K (Thr389) within these resistant cell lines was also further confirmed by western blotting of independent samples (*Figure 2C*).

The clear separation of resistant and sensitive cell lines within this multi-dimensional analysis suggests an antagonistic relationship between apoptosis promoted through the progressive accumulation of DNA-damage following a cisplatin pulse, and elevated mTOR, MAPK, Akt, STAT or NF-κB signalling events. However, as a statistical process, relationships derived from a principal component analysis are purely correlative in nature and require a further degree of validation before any causative conclusions may be drawn. We therefore sought to identify effectors of platinum resistance by including specific inhibitors of these signalling pathway components during and after the 2 hr cisplatin pulse, followed by a measurement of apoptosis at 72 hr (*Figure 3B*). Using the resistant A549 (*Figure 3C*) and NCI-H358 (*Figure 3D*) cell lines we observed that specific inhibitors of Akt (MK2206), STAT3 (S3I-201), MEK (UO126) or NF-κB (SC-75741) did not significantly increase apoptosis in either cell line, suggesting that while elevated activity of these signalling proteins may be present in one or both of these resistant lines (*Figure 2C*), they are not causally associated with resistance to cisplatin. Instead, under these conditions, only the inhibition of P70S6K with the dual PI3K/mTOR inhibitor dactolisib resulted in a significant increase in cisplatin-induced apoptosis in both cell lines. P70S6K is a serine/threonine-specific protein kinase known to require phosphorylation by both PI3K and mTOR for activation (*Sunami et al., 2010*; *Moser et al., 1997*). While dactolisib will result in the inhibition of a number of substrates downstream of both PI3K and mTOR, the lack of sensitisation by an Akt inhibitor (MK2206) suggests a specific role for P70S6K in mediating resistance to cisplatin.

Taking the opposite approach, elevated levels of JNK pathway activity (pJNK, pc-Jun) were also observed following cisplatin treatment within the SW-1573 line (*Figure 2C,D,E*). However, the inclusion of a JNK inhibitor (JNK inhibitor VIII) did not prevent cisplatin induced apoptosis in this cell line (*Figure 2—figure supplement 1C*), suggesting that JNK activity was not promoting apoptosis in this context.

## P70S6K promotes platinum resistance in lung adenocarcinoma

Comparing the relative expression levels of phosphorylated P70S6K across our stratified panel of cell lines, higher expression was observed in the resistant NCI-H358 and NCI-H1573 lines (*Figure 2C*). This expression pattern was also mirrored by the levels of total P70S6K (*Figure 2C*), which seems to be primarily driving the observed levels of P70S6K phosphorylation. Interestingly, the resistant A549 line did not have elevated P70S6K expression or phosphorylation, although as reflected in the PCA plot (*Figure 2E*) this cell line did have high expression of the P70S6K substrate, Ribosomal Protein S6, leading to a level of phosphorylation equivalent to that observed in the resistant NCI-H358 line (*Figure 2C*).

Utilising the highest expressing (NCI-H358) and lowest expressing (NCI-H1299) lines, which are also both *TP53* null, we observed that cisplatin treatment resulted in greater caspase 3 cleavage and γH2A.X expression in the sensitive NCI-H1299 line (*Figure 3E*), which is in line with the multiplexed

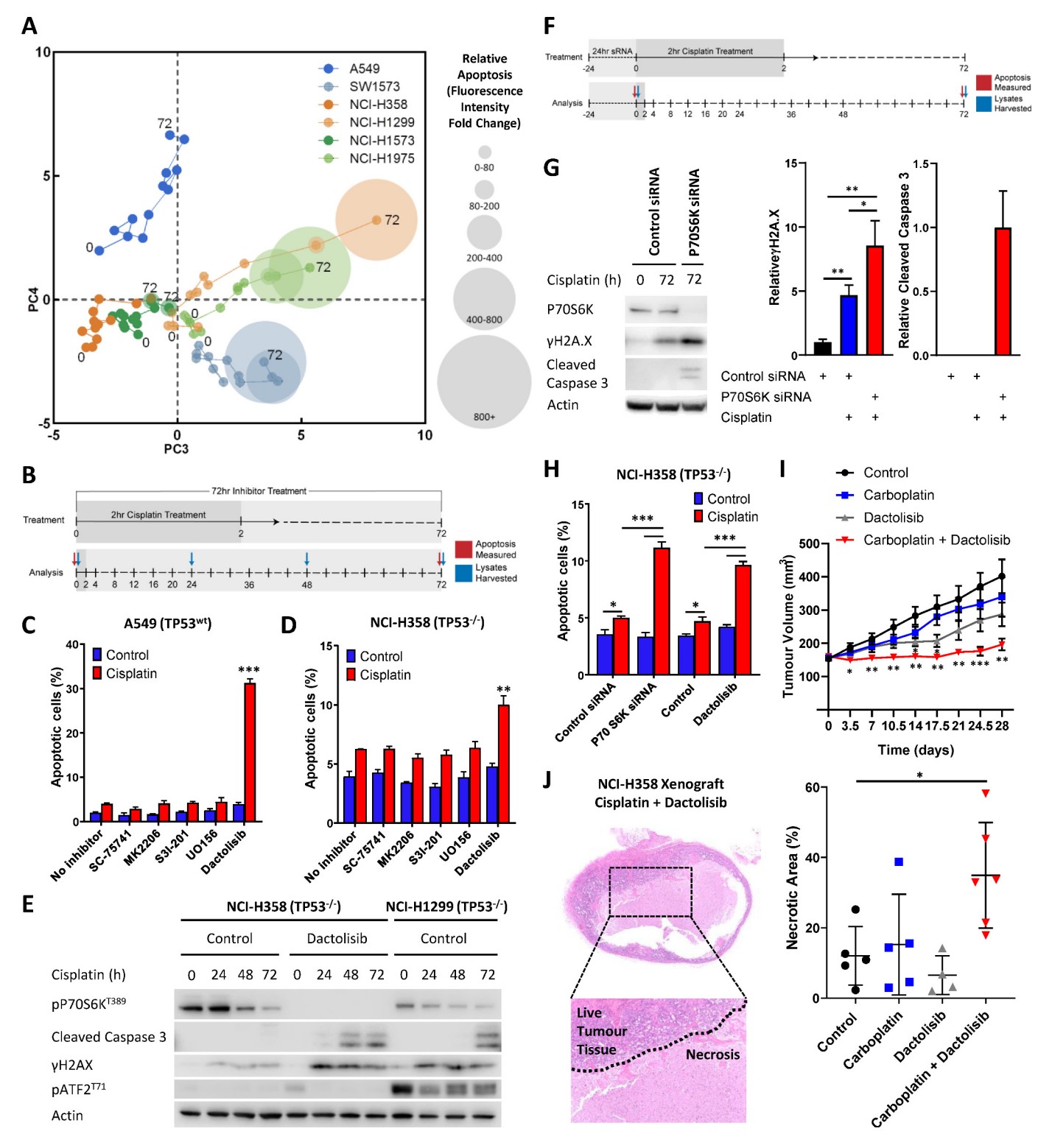

**Figure 3.** Visualising and targeting mechanisms of cisplatin resistance. (**A**) An overlay of real-time imaging of apoptosis following a cisplatin pulse, performed on the Incucyte platform using a fluorescent caspase substrate (1 μM), on top of the PCA plot of component 3 against component 4. (**B**) Schematic of the cisplatin pulse model, with the addition of small molecule inhibitors, and outline of sample collection for apoptosis assays and western blotting. (**C,D**) Apoptosis measured by propidium iodide staining for the sub-G1 population in A549 and NCI-H358 cells, performed 72 hr following a cisplatin pulse with the addition of small molecule inhibitors (1 μM) as indicated (n = 3, mean ± SD). Statistical significance was determined by t-test

*Figure 3 continued on next page*

Figure 3 continued

(***p<0.001, **p<0.01). (E) Western blotting on lysates from NCI-H358 and NCI-H1299 cells following a cisplatin pulse, with the addition of dactolisib (1 µM), as indicated. (F) Schematic of the cisplatin pulse model, with the addition of siRNA pre-treatment, and outline of sample collection for apoptosis assays and western blotting. (G) Western blotting on lysates from NCI-H358 cells, treated with P70S6K or control siRNA, as indicated, prior to and following a cisplatin pulse (n = 3, mean ± SD). Statistical significance was determined by one-way ANOVA (**p<0.01, *p<0.05). (H) Apoptosis measured by propidium iodide staining for the sub-G1 population in NCI-H358 cells, treated with P70S6K, control siRNA or dactolisib, as indicated, performed 72 hr following a cisplatin pulse (n = 3, mean ± SD). Statistical significance was determined by one-way ANOVA (***p<0.001, *p<0.05). (I) Tumour growth in nude mice bearing NCI-H358 xenografts with continuous treatment of vehicle control or dactolisib (45 mg/kg) prior to, and following a single dose of carboplatin (60 mg/kg) (n ≥ 4, mean ± SEM). Statistical significance was determined by one-way ANOVA at each time point (***p<0.001, **p<0.01, *p<0.05). (J) Quantification of necrosis in NCI-H358 xenografts following the treatment described in (I) (n ≥ 4, mean ± SD). Statistical significance was determined by one-way ANOVA *p<0.05.

The online version of this article includes the following figure supplement(s) for figure 3:

**Figure supplement 1.** P70S6K knockdown in NCI-H358 and NCI-H1299 cells.

signalling analysis (*Figure 1D*). Crucially, treatment of the resistant NCI-H358 line with dactolisib during and after the cisplatin pulse increased both caspase 3 cleavage and γH2A.X expression to that observed in the sensitive NCI-H1299 line. Conceptually, this now mimics the movement of the resistant NCI-H358 line along PC3 of our PCA analysis, towards the region of DNA damage and apoptosis characterised by the sensitive cell lines (*Figures 2* and *3A*). This finding demonstrates that while this form of multi-dimensional analysis of signalling networks creates a set of correlative relationships, this data can also be utilised to investigate causal effectors of downstream cellular behaviour.

As mentioned above, dactolisib can efficiently inhibit the phosphorylation of P70S6K, but will also result in the inhibition of a number of other potential PI3K/mTOR substrates. To confirm that P70S6K is a key downstream component mediating platinum resistance in this context, we knocked down P70S6K using two independent siRNAs in the NCI-H358 line, prior to proceeding with a cisplatin pulse (*Figure 3F*, *Figure 3—figure supplement 1A*). Under these conditions, cisplatin treatment resulted in significantly higher caspase 3 cleavage and γH2A.X expression in the P70S6K knockdown NCI-H358 cells (*Figure 3G*), whilst P70S6K knockdown did not further sensitise the already cisplatin-sensitive NCI-H1299 cell line (*Figure 3—figure supplement 1B*). Additionally, P70S6K knockdown with both siRNAs also significantly increased cisplatin-induced apoptosis in the NCI-H358 cells, to the same extent as that observed with dactolisib treatment (*Figure 3H*, *Figure 3—figure supplement 1C*).

To confirm the efficacy this combination therapy approach in vivo and validate the findings of our pulse model, we treated mice bearing NCI-H358 xenografts with either carboplatin, dactolisib, or a combination of both (*Figure 3I*). For this model, dactolisib (45 mg/kg) was delivered by oral gavage, prior to a one-off intraperitoneal injection of carboplatin (60 mg/kg), and throughout the course of the experiment. Under these conditions, carboplatin had no significant effect upon tumour growth, while dactolisib had a moderate effect as a single agent that was only significant at days 14 and 17. However the combination of carboplatin and dactolisib completely halted tumour growth, even up to 28 days following the single pulse of carboplatin treatment. Furthermore, an analysis of tumour sections revealed that the carboplatin and dactolisib treated xenografts were not only smaller in size, but also had significantly larger regions of necrosis (*Figure 3J*).

These findings demonstrate that elevated P70S6K activity can specifically mediate resistance to platinum-based chemotherapy in lung adenocarcinoma, which can be effectively targeted with the dual PI3K/mTOR inhibitor dactolisib. Importantly, elevated P70S6K activity is frequently observed in several cancer subtypes, including lung cancer (*Chen et al., 2017*), and its overexpression is commonly associated with aggressive malignant phenotypes and poor overall prognoses (*Ip and Wong, 2012*). Indeed, an analysis of TCGA data using cBioPortal (*Gao et al., 2013*; *Cerami et al., 2012*; *Hoadley et al., 2018*) reveals that *RPS6KB1* and *RPS6KB2*, the two isoforms of P70S6K, are amplified or over-expressed in 20% and 11% of lung adenocarcinoma cases, respectively (*Figure 4A*). Importantly, while *TP53* mutations or deletions also occur within 47% of lung adenocarcinomas, *RPS6KB1* and *RPS6KB2* amplification/over-expression occur on the background of either wildtype or mutant *TP53*.

Further analysis with KM plotter (*Győrffy et al., 2013*) demonstrated that the elevated mRNA expression of both isoforms was also significantly associated with poor overall survival of lung

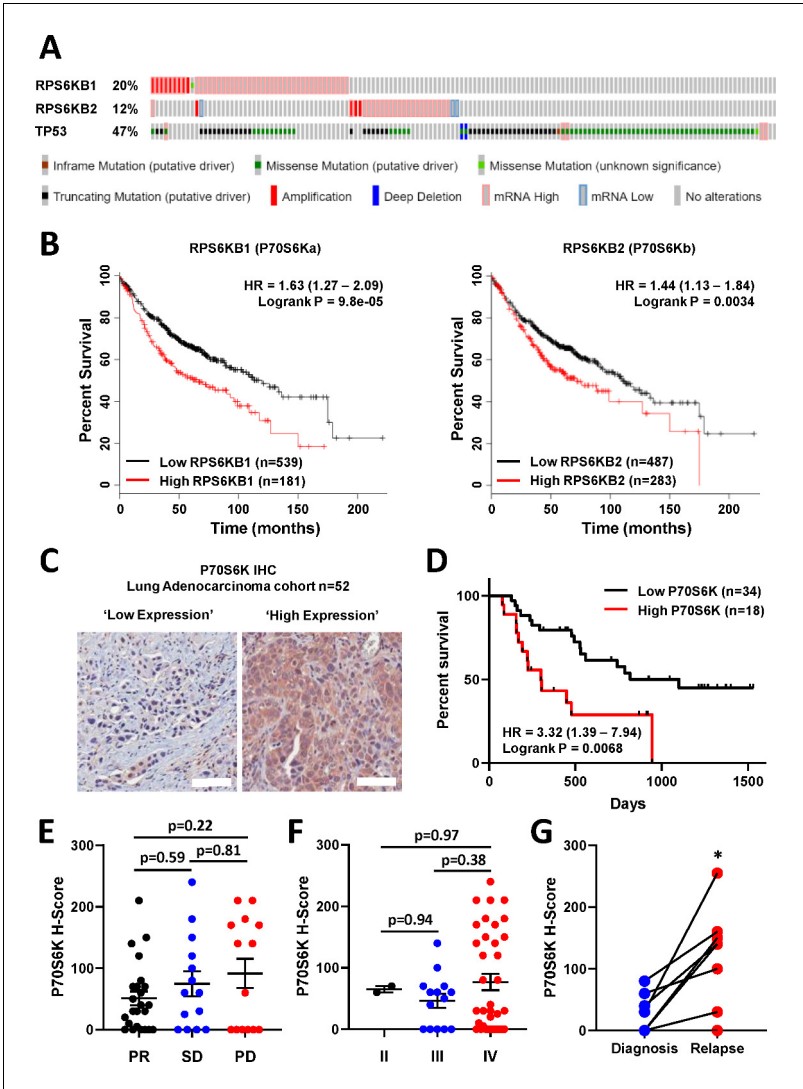

**Figure 4.** P70S6K in lung adenocarcinoma. (**A**) The frequency of somatic mutations and mRNA over-expression for the *RPS6KB1*, *RPS6KB2* and *TP53* genes in a publically available cohort of lung adenocarcinoma patients (cBioPortal). (**B**) The association between *RPS6KB1* and *RPS6KB2* mRNA expression and overall patient survival in a publically available patient cohort (KM Plotter). Statistical significance was determined by log rank test. (**C**) Representative images of P70S6K immuno-histochemistry staining from a cohort of 52 lung adenocarcinoma patients (Scale bar = 100 µM). (**D**) Survival analysis based upon P70S6K staining in this cohort. Statistical significance was determined by log rank test. (**E**) The association between tumour P70S6K staining intensity (H-Score) and patients that underwent a partial response (PR), or presented with stable disease (SD) or progressive disease (PD). Statistical significance was determined by one-way ANOVA, (**F**) The association between tumour P70S6K staining intensity and tumour stage. Statistical significance was determined by one-way ANOVA. (**G**) P70S6K staining in eight matched patient samples from diagnosis and relapse (n = 8). Statistical significance was determined by t-test (*p<0.05).

adenocarcinoma patients (*Figure 4B*). We therefore investigated the association with survival and response to chemotherapy by performing IHC for total P70S6K in a cohort of 52 lung adenocarcinoma patients that all received a neoadjuvant chemotherapy regimen containing platinum-based chemotherapy (*Figure 4C*, *Supplementary file 4*). In this cohort, high expression of P70S6K was significantly associated with poor overall survival (*Figure 4D*), while there was a non-significant trend towards higher P70S6K expression in patients with progressive disease (*Figure 4E*) and a later disease stage (*Figure 4F*). In eight patients with matched diagnosis and relapse samples, the levels of

P70S6K expression were also significantly increased upon relapse (*Figure 4G*), further highlighting the functional role of P70S6K in the cellular response to platinum therapy.

## P70S6K promotes cell cycle arrest in response to cisplatin

As P70S6K has a known role in cell cycle progression (*Lane et al., 1993*), we performed live cell imaging of the FUCCI two-colour sensor of cell cycle progression (*Sakaue-Sawano et al., 2008*) across the resistant A549, NCI-H1573 and NCI-H358 cell lines. This approach allowed us to track the cell cycle progression and fate of individual cells for 72 hr following treatment with a 2 hr pulse of cisplatin (*Figure 5A*). In this assay, treatment with a single pulse of cisplatin caused several notable cell cycle responses. First, A549 (*TP53* wildtype) and NCI-H358 (*TP53* null) cells were significantly more likely to remain arrested in G1, both before (G1 arrest before mitosis; ABM) and after undergoing mitosis (G1 arrest after mitosis; AAM) (*Figure 5B,C*). In contrast, NCI-H1573 cells did not show any significant increase in G1 transit time, as may be expected from a *TP53* mutant cell line.

For all cell lines, cisplatin pulsing resulted in a significant increase in S/G2 transit time (*Figure 5B, C*). Notably, the NCI-H1573 (*TP53* mutant) and NCI-H358 (*TP53* null) cells delayed for more time in G2 (G2 arrest; mean 2443 min and 2555 min respectively) compared to A549 cells (mean 1426 min), after which the mutant and null cells often entered a prolonged aberrant mitosis (*Figure 5—figure supplement 1*), resulting in a small increase in cells dying during mitosis or in the following G1/S phase (*Figure 5C*, death after mitosis; DAM). In contrast, A549 cells rarely entered into mitosis, instead many of these S/G2 arrested cells turned from green (S/G2) back to red (G1) without undergoing mitosis (G2-exit) (*Figure 5—figure supplement 1*). A similar G2-exit, senescence state has been reported to be dependent on p21 (*Gire and Dulic, 2015*) and likely provides *TP53* wildtype cells protection from death by preventing progression through an aberrant mitosis. Therefore, to confirm the specific role of p53 in this context, this assay was repeated with an siRNA-mediated knockdown of p53 in A549 cells (*Figure 5—figure supplement 2*). In line with this hypothesis, this orthogonal approach revealed a significant increase in G2-arrest in the p53 knockdown cells, along with a decrease in occurrence of G2-exit, effectively pheno-copying the observed difference between the A549 wildtype and NCI-H358 p53 null cell lines (*Figure 5C*).

Treatment of all cell lines with dactolisib as a single agent also caused a significant increase in G1 arrest both before (G1 ABM) and after mitosis (G1 AAM), irrespective of *TP53* status (*Figure 5B,C*). Consequently, for all cell lines, dactolisib treatment of cells within early G1 phase during the cisplatin pulse resulted in many cells remaining arrested and viable in G1 (G1 ABM), highlighting the importance of DNA replication for inducing cisplatin induced toxicity and killing. However, for the *TP53* wildtype and null cells that were in late G1 or S phase at the time of cisplatin pulsing, dactolisib treatment significantly reduced S/G2 transit time (*Figure 5B*) and greatly increased the number of cells that underwent apoptosis before mitosis (Death before mitosis; DBM) (*Figure 5C,D*). Notably, dactolisib treatment did not prevent cisplatin treated cells from entering into an aberrant mitosis, but did significantly increase the duration of mitotic arrest, independent of *TP53* status (*Figure 5B*), which correlated with an increase in death during or after mitosis (DAM) in *TP53* wildtype and null, but not mutant cells (*Figure 5C,D*). Surprisingly, dactolisib treatment did not result in any sensitisation of *TP53* mutant H1573 cells to cisplatin (*Figure 5D*), which also corresponded with a failure of dactolisib to significantly reduce S/G2 phase transit time (*Figure 5B*).

In summary, combination therapy with dactolisib sensitised actively cycling cells to cisplatin through distinct mechanisms dependent on *TP53* status. In the *TP53* wildtype A549 cells it induced pre-mitotic cell death and prevented cells from undergoing a protective G2-exit, which is likely partially dependent on the p53-p21 axis. Similarly, in the *TP53* null line, dactolisib promoted both pre-mitotic cell death and the entry of cells into a prolonged deleterious aberrant mitosis, the latter likely due to a weakened p53-dependent G2 checkpoint (*Engeland, 2018*). While in *TP53* mutant H1573 cells, there was no significant sensitisation due to a sustained S/G2 arrest in the presence of dactolisib.

To further confirm the specific role of P70S6K in this cell cycle phenotype, this assay was repeated along with siRNA mediated P70S6K knockdown in the resistant NCI-H358 cell line (*Figure 5—figure supplement 3*). This approach demonstrated that specific ablation of P70S6K in this model resulted in a non-significant trend towards a shortened G2-arrest with both P70S6K siRNAs, and a greatly increased cell death both before (DBM) or after mitosis (DAM). While the non-significant decrease in G2-arrest observed upon siRNA-mediated knockdown of P70S6K may have been limited by

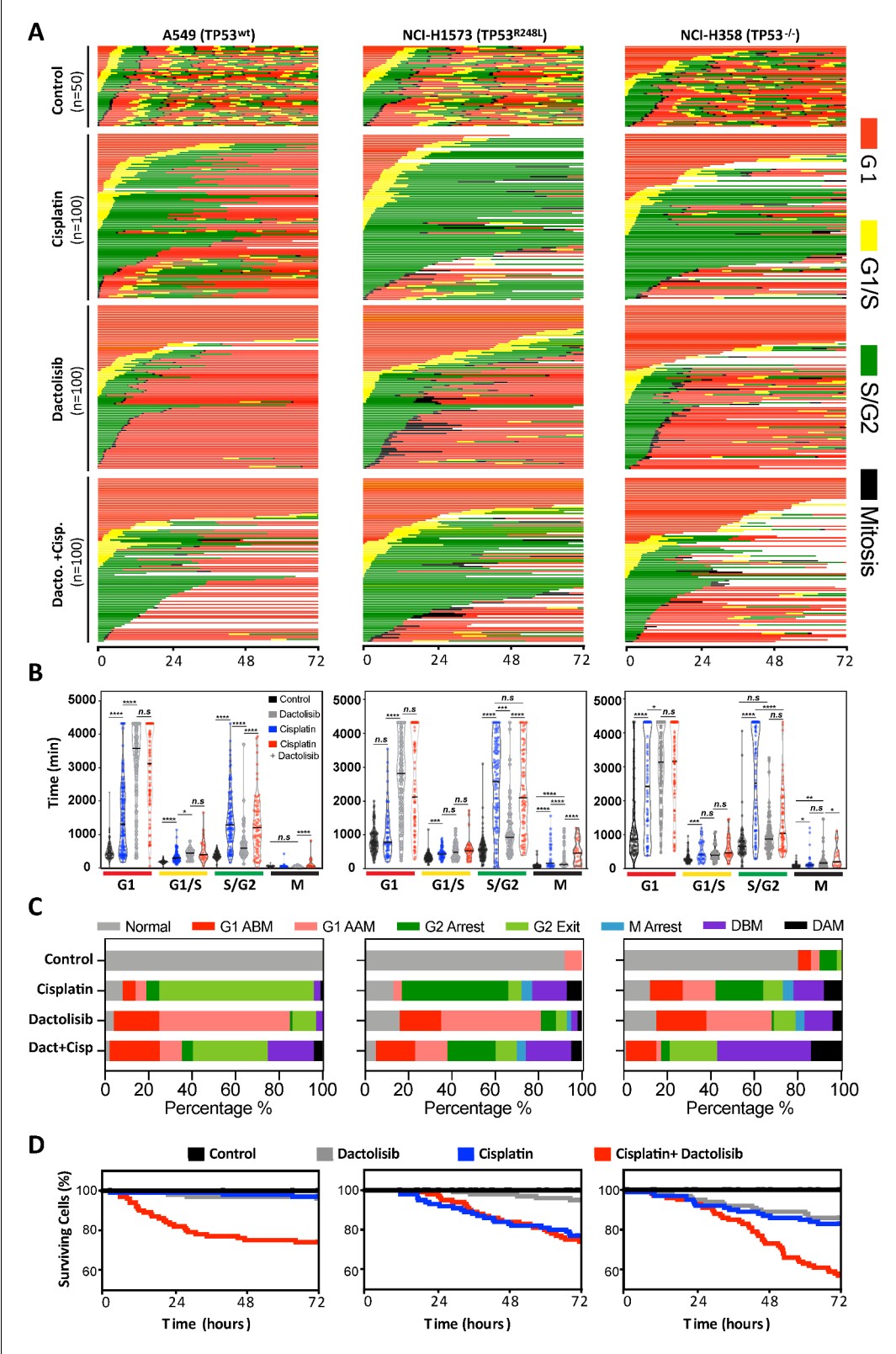

**Figure 5.** FUCCI analysis of cell cycle progression. (**A**) Live-cell imaging of the FUCCI biosensor proteins mVenus-hGeminin(1/110) and mCherry-hCdt1 (30/120), stably expressed by the A549, NCI-H1573 and NCI-H358 cell lines. Images were taken every 20 min for 72 hr under control conditions, or following a cisplatin pulse (5 μg/mL, 2 hr) in the presence or absence of dactolisib (1 μM). (**B**) Quantification of the length of each cell cycle phase under each treatment condition (n = 17–175, mean ± SD). Statistical significance was determined by one-way ANOVA (****p<0.0001, ***p<0.001, **p<0.01,

*Figure 5 continued on next page*

*Figure 5 continued*

*p<0.05. (**C**) Quantification of fate of each cell; including G1 arrest before mitosis (G1 ABM), G1 arrest after mitosis (G1 AAM), death before mitosis (DBM) and death after mitosis (DAM). (**D**) Survival curves indicating the proportion of viable cells over time under each treatment condition.

The online version of this article includes the following figure supplement(s) for figure 5:

**Figure supplement 1.** Representative images of cells expressing the FUCCI biosensor, undergoing an aberrant mitosis, G2 exit, Death before Mitosis (DBM) and Death after Mitosis (DAM) following a cisplatin pulse (5 µg/mL, 2 hr).

**Figure supplement 2.** FUCCI analysis following p53 knockdown.

**Figure supplement 3.** FUCCI analysis following P70S6K knockdown.

---

potential heterogeneity in the level of knockdown achieved, the increase in cell death further suggests that dactolisib mediated inhibition of P70S6K is responsible for the sensitisation of lung adenocarcinoma cells to cisplatin treatment.

## Response to dactolisib is dependent upon TP53 status

A notable observation arising from this cell cycle analysis was the ability of dactolisib to increase apoptosis by preventing an unexpected cell cycle arrest in the *TP53* null NCI-H358 cell line. Interestingly, this association between cell cycle arrest and cisplatin resistance can also be observed from the original analysis of signalling dynamics (*Figure 1D*, *Figure 1—figure supplement 4*). While both the NCI-H1299 and NCI-H358 cell lines are *TP53* null, there was an increased expression of p21 following cisplatin treatment in the resistant NCI-H358 line, but not the sensitive NCI-H1299 line. Importantly, the increased p21 expression by NCI-H358 cells was also observed by western blotting of independent samples and was efficiently inhibited by both treatment with dactolisib (*Figure 6A, B*) and the specific knockdown of P70S6K with siRNA (*Figure 6C*). Taken together, this data suggests that P70S6K activity is necessary for promoting p21 expression in order to maintain cell cycle arrest following cisplatin treatment. However, in the absence of p53, we observed that this may potentially be mediated by the related transcription factor p63, which is elevated in the NCI-H358 line following treatment with cisplatin, and also inhibited by the addition of dactolisib (*Figure 6A,B*) or treatment with P70S6K siRNA (*Figure 6C*).

Given the potential role for p21 in mediating resistance to cisplatin in the NCI-H358 line, we investigated whether sensitisation by dactolisib would be generalizable across the original panel of 6 lung adenocarcinoma cell lines with differing *TP53* status. In accordance with this hypothesis, and in line with the FUCCI single-cell imaging (*Figure 5A*), both the *TP53* wildtype cell lines were significantly sensitised to cisplatin by the addition dactolisib, as was the resistant *TP53* null line, NCI-H358 (*Figures 2D, H* and *6D*). The sensitive *TP53* null line, NCI-H1299, was not further sensitised and appeared to already be at the upper limit of apoptosis within this model. Also, in line with the FUCCI analysis, dactolisib was not able to increase cisplatin-induced apoptosis in the *TP53* mutant line NCI-H1573, whilst it significantly antagonised cisplatin-induced apoptosis in the *TP53* mutant NCI-H1975 line.

While *TP53* mutation status alone is not sufficient to determine sensitivity to cisplatin (*Figure 1B*), this dependence upon *TP53* mutation status for dactolisib-mediated sensitisation to cisplatin treatment is likely explained by considering the ability of p21 to either maintain a G2-arrest, or promote an aberrant G2-exit (*Gire and Dulic, 2015*), thereby preventing DNA damage induced apoptosis occurring during, or shortly after mitosis. In line with this hypothesis, the addition of dactolisib to the *TP53* wildtype A549 cell line inhibited cisplatin-induced p53 and p21 accumulation, resulting in elevated markers of DNA damage (γH2A.X expression) and apoptosis (caspase 3 cleavage) following cisplatin treatment (*Figure 6E*). However, in the *TP53* mutant NCI-H1573 line, which was not sensitised by dactolisib, p21 expression was not significantly elevated following cisplatin treatment, and γH2A.X expression and caspase 3 cleavage were unchanged by the addition of dactolisib (*Figure 7*).

To further validate this link between sensitisation by dactolisib and *TP53* mutation status, we took another panel of 8 lung adenocarcinoma cell lines (*Supplementary file 5*) and correlated their relative expression of phosphorylated P70S6K (*Figure 6F*) to their apoptotic response to cisplatin (*Figure 6G*). This secondary analysis revealed a general trend towards higher levels of phosphorylated P70S6K and decreased apoptosis in response to cisplatin across the whole panel. However, the strength of this correlation was greatly increased when considering the *TP53* status of the lines,

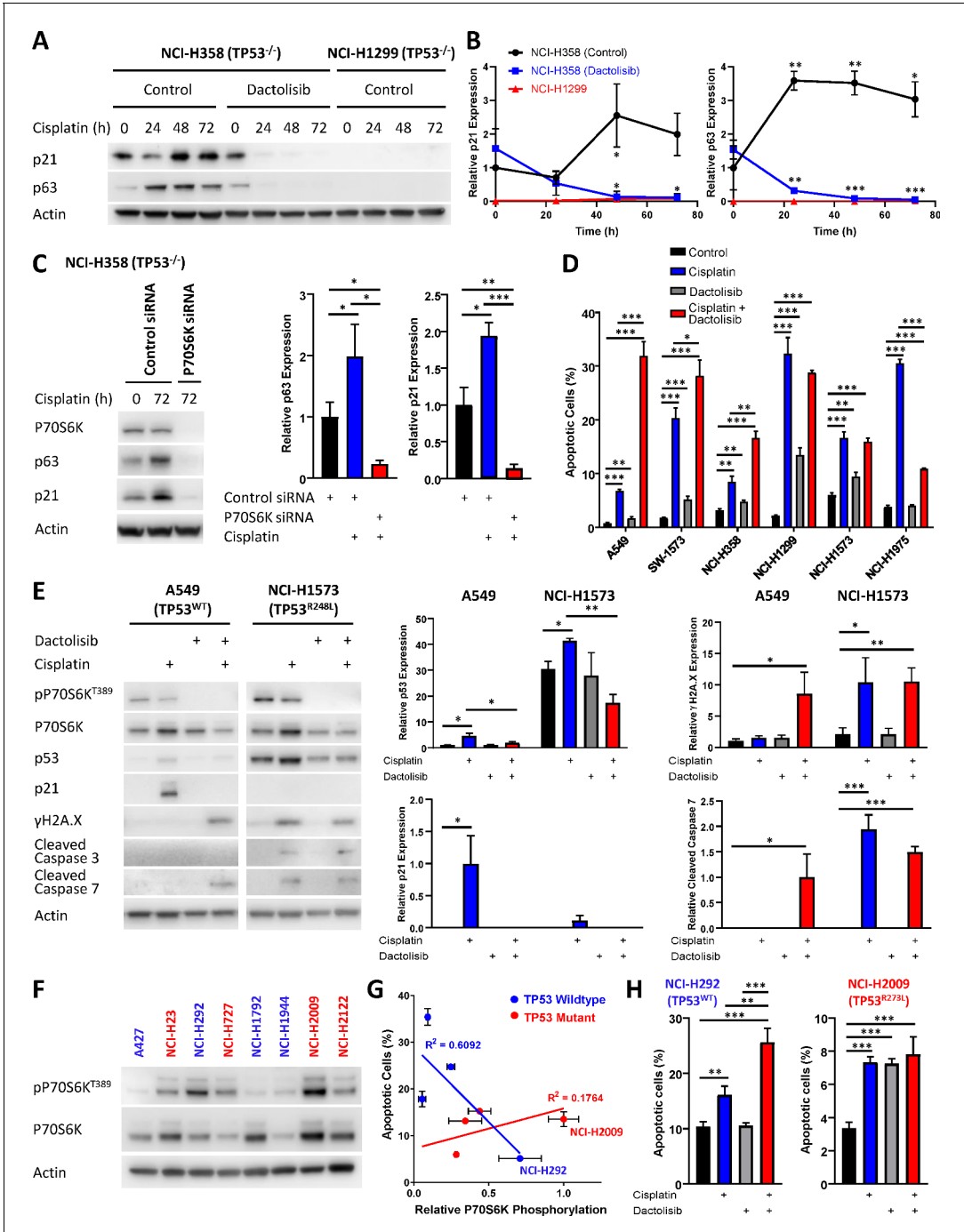

**Figure 6.** Response to dactolisib is dependent upon *TP53* mutation status. (**A**) Western blotting on lysates from NCI-H358 and NCI-H1299 cells following a cisplatin pulse, with the addition of dactolisib (1 μM), as indicated. (**B**) Quantification of effects of dactolisib upon cisplatin induced p63 and p21 expression (n = 3, mean ± SD). (**C**) Western blotting on lysates from NCI-H358 cells, treated with P70S6K or control siRNA, as indicated, prior to and following a cisplatin pulse (n = 3, mean ± SD). (**D**) Apoptosis measured by propidium iodide staining for the sub-G1 population performed 72 hr following a cisplatin pulse with the addition of dactolisib (1 μM) as indicated (n = 3, mean ± SD). (**E**) Western blotting on lysates from A549 and NCI-H1573 cells, 48 hr following a cisplatin pulse, with the addition of dactolisib (1 μM), as indicated (n = 3, mean ± SD). (**F**) Western blotting across a second panel of lung adenocarcinoma cell lines. (**G**) Correlation between P70S6K phosphorylation and apoptosis, as measured by propidium iodide staining for the sub-G1 population, performed 72 hr following a cisplatin pulse (n = 3, mean ± SD). For all panels the statistical significance was determined by one-way ANOVA (***p<0.001, **p<0.01, *p<0.05).

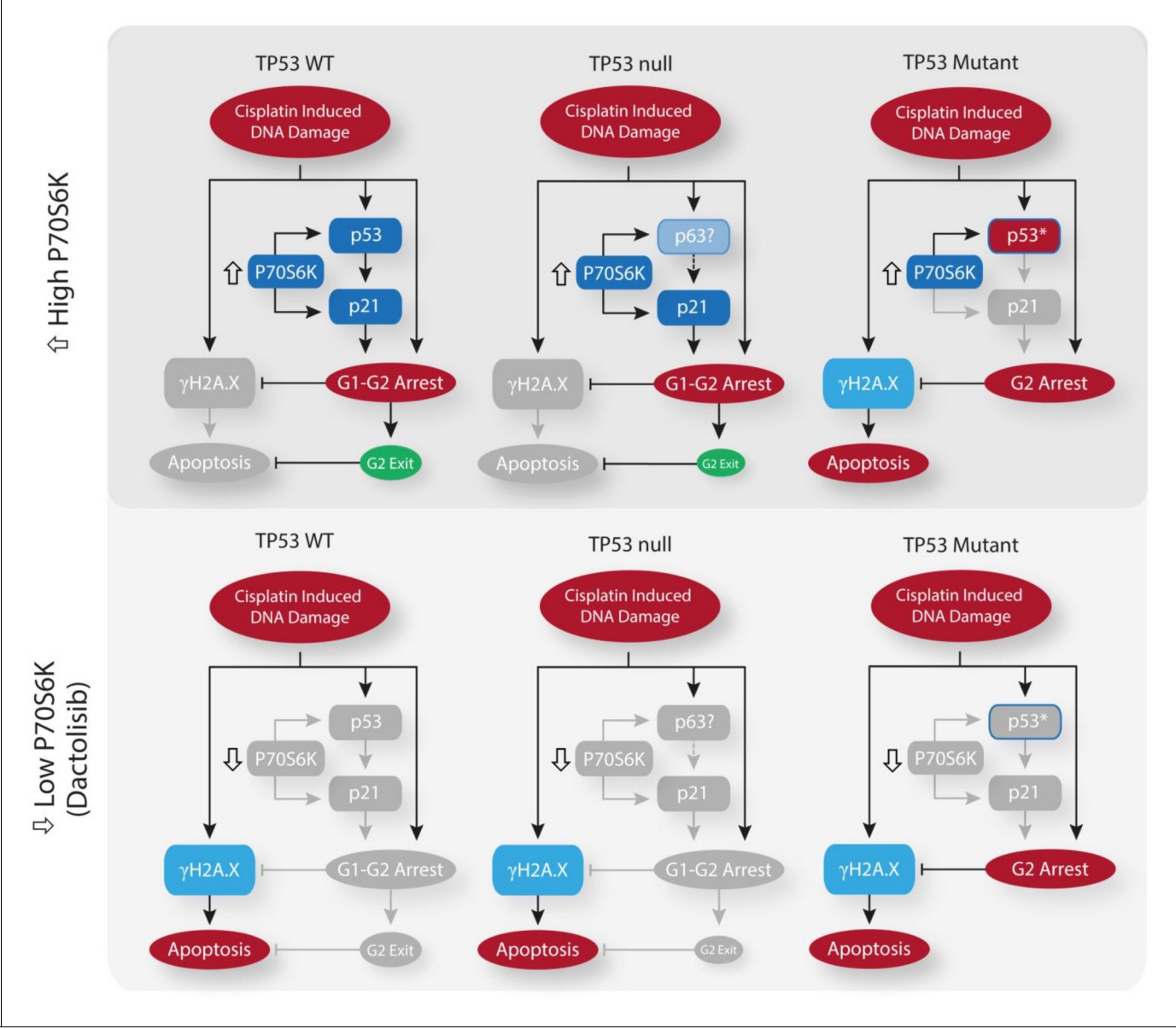

**Figure 7.** Schematic outlining the effect of cisplatin on cell cycle arrest, DNA damage and apoptosis across the spectrum of TP53 mutation states, and the influence of P70S6K expression levels or inhibition upon these processes.

with a strong correlation observed for wildtype *TP53* lines ($r^2$ = 0.6092) but no correlation for *TP53* mutant lines ($r^2$ = 0.1764). In agreement with our finding that elevated P70S6K activity promotes resistance in a *TP53* dependent manner, combination therapy with cisplatin and dactolisib significantly sensitised the high pP70S6K/*TP53* wildtype NCI-H292 cell line, but not the high pP70S6K/*TP53* mutant NCI-H2009 line. Therefore, this data demonstrates that P70S6K promotes an aberrant G2-exit or prolonged G2-arrest following cisplatin treatment, presumably through enhanced translation of p53/p21 in *TP53* wildtype cell lines and p63/p21 in *TP53* null lines (*Figure 7*). However, in the presence of mutant p53, P70S6K is not able to influence this DNA damage induced signalling axis and is therefore not associated with sensitivity to cisplatin.

## Discussion

A number of studies have been performed investigating the potential mechanisms of resistance to platinum-based chemotherapies (*Stewart, 2007*), although this has yet to result in the identification of clinically successful combination therapies for lung adenocarcinoma. One potential explanation for the high number of in vitro findings that have not translated to the clinic is the use of experimental techniques that do not replicate the in vivo pharmacokinetics of platinum therapies (*Shen et al., 2012*). To date, most in vitro approaches have involved continuously culturing cell lines in the presence of high doses of platinum chemotherapy for multiple days, potentially allowing for more extensive DNA damage and off-target effects that would not be seen in vivo. Here we have utilised a pulse model that more accurately models both the concentration and timing of cisplatin that would be observed clinically. This is an important consideration, as platinum therapies are known to act via the formation of DNA adducts, although they are also capable of bonding with proteins and RNA (*Jamieson and Lippard, 1999*). It is likely that the continuous culturing of cells in the presence of high doses of these drugs in vitro would result in the accumulation of numerous off-target adducts and the activation of a stress response incongruent with the actual in vivo apoptotic mechanism. Indeed, this effect may be apparent in *Figure 1—figure supplement 1D*/2, where p38 activity is induced by the continuous exposure model, but not by the pulse model. A number of previous studies have implicated the activity of p38 in mediating resistance to cisplatin (*Sarin et al., 2018*; *Pereira et al., 2013*; *Galan-Moya et al., 2011*; *Hernández Losa et al., 2003*), however we did not observe highly elevated p38 signalling in any cell line following the cisplatin pulse (*Figure 1D*), nor a correlation between p38 activity and the apoptotic response (*Figure 2*).

Instead, through the use of this pulse model, and by overlaying real-time apoptosis data onto a multivariate signalling analysis across a panel of lung adenocarcinoma lines, we have now identified elevated P70S6K activity as an effector of inherent platinum chemoresistance. High levels of P70S6K expression have previously been associated with aggressive tumour behaviour in lung cancer (*Chen et al., 2017*), as well as other cancer types such as breast (*Holz, 2012*), colorectal (*Nozawa et al., 2007*) and liver (*Sahin et al., 2004*) cancers. This is also supported by our analysis of P70S6K expression patient samples (*Figure 4*), where high mRNA and protein levels were both associated with poor overall survival, and P70S6K protein expression was significantly elevated in relapsed tumours. Importantly, we also propose that combination therapy with the dual PI3K/mTOR inhibitor dactolisib can sensitise lung adenocarcinomas to platinum chemotherapy, although only in the context *TP53* WT or *TP53* null tumours. This detailed mechanistic understanding is especially important given that P70S6K amplification can occur on a background of either mutant or wildtype *TP53* (*Figure 4A*).

Functionally, dactolisib will result in the inhibition of a number of substrates downstream of PI3K and mTOR. However, as the specific knockdown of P70S6K with siRNA recapitulated the effects of dactolisib on p63, p21, γH2A.X, cleaved caspase 3 and apoptosis, it is likely that the elevated expression and activity of P70S6K in resistant lung adenocarcinoma cells is the functional target of dactolisib and a causal effector of platinum resistance in these cells. Further supporting this central role of P70S6K, previous research has demonstrated that P70S6K can regulate cell cycle progression via the enhanced translation of p21 mRNA (*Guégan et al., 2014*), although our data also suggest that P70S6K may also play a role in the enhanced expression of both p53 and p63 in this context.

The combination of cisplatin and dactolisib has also previously been proposed for osteosarcoma (*Huang et al., 2018*), head and neck squamous cell carcinoma (*Hsu et al., 2018*) and bladder cancer (*Moon et al., 2014*). However, our finding that this combination therapy is effective in *TP53* WT and null, but not *TP53* mutant tumours, has significant implications for the clinical application of these drugs and the design of potential clinical trials. This fine grained stratification of response is an important finding, as the failure of many clinical trials is frequently attributed to the lack of suitable biomarkers to stratify the patient cohort (*Hay et al., 2014*), highlighting the value of our approach that is capable of dissecting the mechanism of variable drug response across differing genetic backgrounds. The inability of dactolisib to regulate p21 expression in *TP53* mutant tumour cells following cisplatin treatment likely underlies the inability of dactolisib to sensitise these cells to cisplatin. Therefore, within the context of a potential clinical trial for this drug combination, the *TP53* mutation status of each patient tumour would be a central consideration for identification of the most effective treatment strategy and an understanding of individual patient response. However, this finding is

also widely applicable to the understanding of how cancer-related signalling networks govern drug response and chemoresistance, and how a thorough characterisation of these dynamics will be necessary for the proper design and implementation of any consistently effective, patient-specific therapy.

# Materials and methods

**Key resources table**

| Reagent type (species) or resource | Designation | Source or reference | Identifiers | Additional information |
|---|---|---|---|---|
| Cell line (*Homo sapiens*) | A549 | ATCC | CCL-185, RRID:CVCL_0023 | |
| Cell line (*Homo sapiens*) | NCI-H358 | ATCC | CRL-5807, RRID:CVCL_1559 | |
| Cell line (*Homo sapiens*) | NCI-H1299 | ATCC | CRL-5803, RRID:CVCL_0060 | |
| Cell line (*Homo sapiens*) | NCI-H1573 | ATCC | CRL-5877, RRID:CVCL_1478 | |
| Cell line (*Homo sapiens*) | NCI-H1975 | ATCC | CRL-5908, RRID:CVCL_1511 | |
| Cell line (*Homo sapiens*) | SW1573 | ATCC | CRL-2170, RRID:CVCL_1720 | |
| Cell line (*Homo sapiens*) | A427 | ATCC | HTB-53, RRID:CVCL_1055 | |
| Cell line (*Homo sapiens*) | NCI-H23 | ATCC | CRL-H5800, RRID:CVCL_1547 | |
| Cell line (*Homo sapiens*) | NCI-H292 | ATCC | CRL-1848, RRID:CVCL_0455 | |
| Cell line (*Homo sapiens*) | NCI-H1944 | ATCC | CRL-5907, RRID:CVCL_1508 | |
| Cell line (*Homo sapiens*) | NCI-H2009 | ATCC | CRL-5911, RRID:CVCL_1514 | |
| Cell line (*Homo sapiens*) | NCI-H2122 | ATCC | CRL-5985, RRID:CVCL_1531 | |
| Antibody | anti-Cisplatin modified DNA antibody (rat monoclonal) | Abcam | ab103261, RRID:AB_10715243 | IF (1:500) |
| Antibody | anti-gamma H2A.X (phospho S139) antibody (mouse monoclonal) | Abcam | Ab26350, RRID:AB_470861 | WB (1:1000) |
| Antibody | Anti-phospho-STAT3 (Ser729) (rabbit monoclonal) | Cell Signaling Technology | #9134, RRID:AB_331589 | WB (1:1000) |
| Antibody | Anti-phospho-Akt (Ser473) (rabbit monoclonal) | Cell Signaling Technology | #9271, RRID:AB_329825 | WB (1:1000) |
| Antibody | Anti-phospho-NF-κB p65 (Ser536) (rabbit monoclonal) | Cell Signaling Technology | #3033, RRID:AB_331284 | WB (1:1000) |
| Antibody | Anti-phospho-p44/42 MAPK (Erk1/2) (Thr202/Tyr204) (rabbit monoclonal) | Cell Signaling Technology | #9101, RRID:AB_331646 | WB (1:1000) |
| Antibody | Anti-phospho-ATF-2 (Thr71) (rabbit monoclonal) | Cell Signaling Technology | #5112, RRID:AB_560873 | WB (1:1000) |

*Continued on next page*

*Continued*

| Reagent type (species) or resource | Designation | Source or reference | Identifiers | Additional information |
|---|---|---|---|---|
| Antibody | Anti-p70 S6 Kinase (rabbit monoclonal) | Cell Signaling Technology | #2708, RRID:AB_390722 | WB (1:1000), IHC (1:600) |
| Antibody | Anti-phospho-p70 S6 Kinase (Thr389) (rabbit monoclonal) | Cell Signaling Technology | #9205, RRID:AB_330945 | WB (1:1000) |
| Antibody | Anti-p21 Waf1/Cip1 (rabbit monoclonal) | Cell Signaling Technology | #2947, RRID:AB_823586 | WB (1:1000) |
| Antibody | Anti-cleaved Caspase-3 (Asp175) (rabbit monoclonal) | Cell Signaling Technology | #9661, RRID:AB_2341188 | WB (1:1000) |
| Antibody | Anti-cleaved Caspase-7 (Asp198) (rabbit monoclonal) | Cell Signaling Technology | #9491, RRID:AB_2068144 | WB (1:1000) |
| Antibody | Anti-p53 (mouse monoclonal) | Santa Cruz Biotechnology | sc-126, RRID:AB_628082 | WB (1:200) |
| Antibody | Anti-p63 (mouse monoclonal) | Novus Biologicals | NB100-691, RRID:AB_10002770 | WB (1:1000) |
| Antibody | Anti-β-Actin (mouse monoclonal) | Sigma Aldrich | AC-15, RRID:AB_476692 | WB (1:5000) |
| Transfected construct (human) | mVenus-hGeminin (1/110) (plasmid) | *Sakaue-Sawano et al., 2008* | | |
| Transfected construct (human) | mCherry-hCdt1 (30/120) (plasmid) | *Sakaue-Sawano et al., 2008* | | |
| Chemical compound, drug | NVP-BEZ235 | Selleck Chem | S1009 | |
| Chemical compound, drug | MK-2206 2HCl | Selleck Chem | S1078 | |
| Chemical compound, drug | S3I-201 | Selleck Chem | S1155 | |
| Chemical compound, drug | U0126-EtOH | Selleck Chem | S1102 | |
| Chemical compound, drug | SC75741 | Selleck Chem | S7273 | |
| Chemical compound, drug | RNaseA | Life Technologies | AM2270 | |
| Chemical compound, drug | Propidium iodide | Sigma-Aldrich | P4170 | |
| Chemical compound, drug | CellTiter96 AQueous Non-Radioactive Cell Proliferation Assay | Promega | G5421 | |
| Chemical compound, drug | Cisplatin | Hospira Australia | 88S035 | |
| Commercial assay or kit | DNA Damage/Genotoxicity Magnetic Bead Panel (7-plex) | Merck Millipore | 48-621MAG | MAGPIX assay, detects ATR (total), Chk1 (Ser345), Chk2 (Thr68), H2A.X (Ser139), MDM2 (total), p21 (total), p53 (Ser15) |
| Commercial assay or kit | TGFβ Signaling Pathway Magnetic Bead Kit (6-plex) | Merck Millipore | 48-614MAG | MAGPIX assay, detects Akt (Ser473), ERK (Thr185/Tyr187), Smad2 (Ser465/Ser467), Smad3 (Ser423/Ser425), Smad4 (total), TGFβRII (total) |

*Continued*

| Reagent type (species) or resource | Designation | Source or reference | Identifiers | Additional information |
|---|---|---|---|---|
| Commercial assay or kit | Bio-Plex Pro Cell Signaling Akt Panel (8-plex) | Bio-Rad | LQ00006JK0K0RR | MAGPIX assay, detects Akt (Ser473), BAD (Ser136), GSK-3α/β (Ser21/Ser9), IRS-1 (Ser636/Ser639), mTOR (Ser2248), p70 S6 kinase (Thr389), PTEN (Ser380), S6 ribosomal protein (Ser235/Ser236) |
| Commercial assay or kit | Bio-Plex Pro Cell Signalling MAPK Panel (9-plex) | Bio-Rad | LQ00000S6KL81S | MAGPIX assay, detects ATF-2 (Thr71), ERK (Thr202/Tyr204, Thr185/Tyr187), HSP27 (Ser78), JNK (Thr183/Tyr185), MEK1 (Ser217/Ser221), p38 MAPK (Thr180/Tyr182), p53 (Ser15), p90 RSK (Ser380), Stat3 (Ser727) |
| Commercial assay or kit | Bio-Plex Pro RBM Apoptosis Panel 2 | Bio-Rad | 171WAR2CK | MAGPIX assay, detects Bad, Bax/Bcl-2 dimer, Bcl-xL, Bim, Mcl-1 |
| Commercial assay or kit | Bio-Plex Pro RBM Apoptosis Panel 3 | Bio-Rad | 171WAR3CK | MAGPIX assay, detects active caspase-3, Bcl-xL/Bak dimer, Mcl-1/Bak dimer, Survivin |
| Commercial assay or kit | Total p53 magnetic bead | Merck Millipore | MP46662MAG | Individual MAGPIX bead kit |
| Commercial assay or kit | Cleaved PARP Magnetic Bead MAPmate | Merck Millipore | 46-656MAG | Individual MAGPIX bead kit |
| Commercial assay or kit | Phospho-NF-κB p65 (Ser536) Set | Bio-Rad | 171V50013M | Individual MAGPIX bead kit |
| Commercial assay or kit | Phospho-IκB-α (Ser32/Ser36) | Bio-Rad | 171V50010M | Individual MAGPIX bead kit |
| Commercial assay or kit | Phospho-c-Jun (Ser63) Set | Bio-Rad | 171V50003M | Individual MAGPIX bead kit |
| Commercial assay or kit | Phospho-Stat1 (Tyr701) Set | Bio-Rad | 171V50020M | Individual MAGPIX bead kit |
| Software | MATLAB and Statistics Toolbox Release 2019a | The Mathworks, Inc | | |
| Commercial assay or kit | NucView 488 Caspase-3 Enzyme substrate | Biotium | #10402 | |
| Other | Matrigel Basement Membrane | Bio-Strategy | BDAA354230 | |
| Chemical compound, drug | carboplatin | Abcam | ab120828 | |
| Antibody | Anti-Ki-67 (rabbit monoclonal) | ThermoFisher scientific | RM-9106, RRID:AB_2335745 | IHC (1:500) |
| Chemical compound, drug | jetPRIME DNA and siRNA transfection reagent | Polyplus transfection | 114–15 | |
| Transfected construct (human) | SignalSilence p70 /85 S6 Kinase siRNA I | Cell Signaling Technology | #6566 | |
| Transfected construct (human) | SignalSilence p70 /85 S6 Kinase siRNA II | Cell Signaling Technology | #6572 | |
| Transfected construct (human) | SignalSilence p53 siRNA I | Cell Signaling Technology | #6231 | |
| Transfected construct (human) | SignalSilence p53 siRNA II | Cell Signaling Technology | #6562 | |

*Continued on next page*

*Continued*

| Reagent type (species) or resource | Designation | Source or reference | Identifiers | Additional information |
|---|---|---|---|---|
| Sequenced-based reagent | SignalSilence Control siRNA (Unconjugated) | Cell Signaling Technology | #6568 | |

## Antibodies, Plasmids, and Reagents

The cisplatin modified DNA (ab103261) and γH2A.X (ab26350) antibodies were from Abcam (Cambridge, USA). The phospho-STAT3 S727 (#9134), phospho-Akt S473 (#9271), phospho-NF-κB S536 (#3033), pERK T202/Y204 (#9101), pATF2 T71 (#5112), (P70S6K (#2708), phospho-P70S6K T389 (#9205), p21 (#2947), cleaved caspase 3 (#9661) and cleaved caspase 7 (#9491) antibodies were from Cell Signaling (MA, United States). The γH2A.X (S139) antibody (AB26350) was from Abcam (MA, USA). The p53 antibody (sc-126) was from Santa Cruz Biotechnology (TX, USA). The p63 antibody (NB100-691) was from Novus Biologicals (CO, USA). The actin monoclonal antibody (AC-15) was from Sigma-Aldrich (MO, United States). The SignalSilence p70/85 S6 Kinase siRNA was from Cell Signaling (MA, United States). The plasmids for FUCCI live cell imaging, mVenus-hGeminin(1/110) and mCherry-hCdt1(30/120), were a kind gift from Dr Atsushi Miyawaki (Riken, Japan). Dactolisib (NZP-BEZ235), MK2206, S3I-201, UO126 and SC75741 were all from Selleck Chem (MA, USA).

## Cell lines

All lung adenocarcinoma cell lines have been previously described (*Marini et al., 2018*). The lines were cultured in Advanced RPMI (12633012, Gibco) containing 1% FCS and 1% GlutaMAX (35050–061, Gibco) under standard tissue culture conditions (5% CO2, 20% O2). All cell lines were authenticated by short tandem repeat polymorphism, single-nucleotide polymorphism, and fingerprint analyses, passaged for less than 6 months. All cell lines were confirmed as negative for mycoplasma contamination using the MycoAlert luminescence detection kit (Lonza, Switzerland).

Stable cell lines expressing the FUCCI biosensor were generated as previously described (*Sakaue-Sawano et al., 2008*) using mVenus-hGeminin(1/110) and mCherry-hCdt1(30/120) probes. Briefly, cells were first transduced with mVenus-hGeminin(1/110) lentiviral particles. Cells were FACS sorted based upon Venus fluorescence, and the resulting cell population transduced again with mCherry-hCdt1(30/120) lentiviral particles. mCherry positive cells were FACS sorted, resulting in a double positive population used for live cell imaging.

## Flow cytometry

Samples for flow cytometry were fixed in −20˚C ethanol overnight, and then resuspended in a DNA staining solution containing 10 mg/mL RNaseA and 1 mg/mL propidium iodide for 30 min before analysis. Flow cytometry was performed using BD FACS Canto II system.

## Confocal microscopy

For the visualisation of cisplatin-DNA adducts following treatment with a pulse of cisplatin (5 μg/mL, 2 hr), cells were grown on Histogrip (Life Technologies) coated glass coverslips. Prior to fixation, cells were treated with 1% Triton X-100 for 1 min, fixed with ice-cold 100% Methanol and stored overnight at −20˚C. The cells were then permeabilized with 0.5% Triton X-100 for 10 min, washed twice with PBS and incubated for 30 min with 2M HCl to denature the DNA. Cells were washed again and incubated in 1% $H_2O_2$ to quench the endogenous peroxidases, before blocking for 30 min and incubated with primary antibody (1:500) overnight at 4 C in blocking solution. The following day, cells were incubated with Secondary-Biotinylated Ab at RT, before incubating them with ABC solution (Vectastain elite ABC HRP kit, Vector Laboratories). Cells were later incubated with TSA solution (TSA plus Cyanine 3 System, PerkinElmer). DNA was stained with H33342 and images collected using a Leica DMI5500 (40x magnification). Images were quantified using Fiji Software. Briefly, color images were split into separate channels. H33342 channel was used to identify the nucleus and generate a mask that was placed on top of the CisPt-DNA channel to quantify the signal coming from the nuclear area. Cytoplasmic area was manually identified in each cell and signal quantified. Background signal was obtained and subtracted from the nuclear and cytoplasm signal. From this data, a nuclear/cytoplasmic ratio was obtained for between 100 and 200 cells at each time point.

## MAGPIX multiplex assays

Multiplex analysis was performed using a Bio-Plex MAGPIX system (#171015044) and Bio-Plex Pro-Wash Station (Biorad). All cell lysates were prepared using standard cell lysis buffer (50 mM Tris HCl pH7.4, 150 mM NaCl, 1 mM EDTA, 1% (v/v) TritonX-100) supplemented with protease and phosphatase inhibitors. Lysates were analysed on all kits, according to manufacturer's instructions. Data was generated using Bio-Plex Manager MP and analysed on the Bio-Plex Manager 6.1 software.

Lysates were analysed using the Milliplex map DNA Damage/Genotoxicity Magnetic Bead Panel (7-plex), Milliplex map Early Apoptosis Magnetic Bead (7-plex), Milliplex map TGFβ Signalling Pathway Magnetic Bead kit (6-plex), Bio-Plex Pro Cell Signalling Akt Panel (8-plex), Bio-Plex Pro Cell Signalling MAPK Panel (9-plex), Bio-Plex Pro RBM Apoptosis Panel two and Bio-Plex Pro RBM Apoptosis Panel 3. Individual beads were also used to analyse NF-κB (Ser536), IκBα (Ser32/36), c-Jun (Ser63), total P53 and cleaved PARP (Biorad). The data was normalized to the median value at the 0 hr time point for each analyte and a log transformation was conducted on the resulting dataset. The principal component analysis was performed using MATLAB and Statistics Toolbox Release 2019a (The Mathworks, Inc, Natick, Massachusetts, United States of America).

## Live-cell imaging of apoptosis

Live-cell imaging of apoptosis was performed using an Essen Bioscience IncuCyte ZOOM Live-Cell Analysis System and a Thermo Fisher Scientific HERAcell 240i $CO_2$ Incubator. Cells were seeded into Corning Costar TC-treated 96-Well Plates and imaged over a 72 hr period at 2 hr intervals over 4 fields of view per well. Caspase activation was visualised using 1 μM NucView 488 Caspase-3 Enzyme Substrate (Biotium). Cell viability was quantified using propidium iodide (1 μg/mL). The generated images were analysed using IncuCyte ZOOM Software Version 2016B. Accompanying cytotoxicity assays were performed using the CellTiter 96 aqueous non-radioactive cell proliferation assay, according to the manufacturer's instructions (Promega).

## Western blotting

Lysates for western blotting were prepared using standard lysis buffer (50 mM Tris HCl pH7.4, 150 mM NaCl, 1 mM EDTA, 1% (v/v) Triton X-100) containing protease inhibitor (p8340, Sigma) and 0.2 mM sodium orthovanadate. The NuPAGE SDS PAGE Gel System and NuPAGE Bis Tris Precast Gels (4–12% and 12%) (Life Technologies) were used to perform gel electrophoresis. Western Lightning PLUS Enhanced Chemiluminescent Substrate (PerkinElmer) was used for imaging western blots on the Vilber Lourmat Fusion chemiluminescent imaging system. Quantitative western blotting was performed using multistrip western blotting, as performed previously (*Kennedy et al., 2019*).

## NCI-H358 xenograft model

NCI-H358 cells ($2 \times 10^6$) resuspended in 100 μL PBS:Matrigel were injected subcutaneously into the flanks of nude mice. Tumour growth was assessed twice weekly by calliper measurement and mice were randomized to treatment arms when tumours reached 150 mm3 (using the formula: width2 x length x 0.5). Carboplatin (60 mg/kg) was delivered by a single tail-vein injection. Dactolisib was prepared in 10% DMSO:90% PEG300 and administered twice-weekly at 45 mg/kg by oral gavage. All in vivo experiments, procedures and endpoints were approved by the Garvan Institute of Medical Research Animal Ethics Committee.

## Immunohistochemistry

Immunohistochemistry was performed on formalin fixed paraffin embedded sections using the Leica BOND RX (Leica, Wetzlar, Germany). Slides were first dewaxed and rehydrated, followed by heat induced antigen retrieval performed with Epitope Retrieval Solution 1 BOND (Leica, Wetzlar, Germany). Primary antibodies were diluted 1:600 (P70S6K) and 1:500 (Ki67) in Leica antibody diluent and incubated for 60 min on slides. Antibody staining was completed using the Bond Polymer Refine IHC protocol and reagents (Leica, Wetzlar, Germany). Slides were counterstained on the Leica Autostainer XL (Leica, Wetzlar, Germany). Leica CV5030 Glass Coverslipper (Leica, Wetzlar, Germany) and brightfield images were taken on the Aperio CS2 Slide Scanner (Leica, Wetzlar, Germany). Quantification of Ki67 staining was performed on three fields of view for each tumour section, and quantified using the particle analysis function of Image J (v1.49).

## FUCCI live-cell imaging

For FUCCI live cell imaging, cells were seeded on 12 well plates and imaged using a Leica DMI6000 using a 20x NA 0.4 objective. Images were taken every 20 min for 72 hr. Before adding any drug (cisplatin, dactolisib or both) an image was taken in order to annotate the cell cycle phase before commencing treatment. Individual cells were tracked manually, with the colour of the nucleus annotated at each time point (Red = G1; Yellow = G1/S, Green = S/G2), the cells were also scored for nuclear envelope breakdown (NEBD) and early signs of anaphase. Mitotic length was calculated by the time period from commencement of NEBD to anaphase. Interphase length was calculated from anaphase to the next daughter cell NEBD. Only one daughter was followed and annotated. Tracking graphs were generated using Prism 7.

## Acknowledgements

The authors would like to acknowledge Dr Atsushi Miyawaki (Riken, Japan) for provision of the mVenus-hGeminin(1/110) and mCherry-hCdt1(30/120) constructs for FUCCI imaging.

## Additional information

### Funding

| Funder | Grant reference number | Author |
|---|---|---|
| Cancer Institute NSW | 2013/FRL102 | David R Croucher |
| Cancer Institute NSW | 15/REG/1-17 | David R Croucher |

The funders had no role in study design, data collection and interpretation, or the decision to submit the work for publication.

### Author contributions

Jordan F Hastings, Conceptualization, Data curation, Formal analysis, Validation, Investigation, Visualization, Methodology; Alvaro Gonzalez Rajal, Conceptualization, Data curation, Formal analysis, Visualization, Methodology; Sharissa L Latham, Investigation, Methodology; Jeremy ZR Han, Formal analysis, Investigation; Rachael A McCloy, Formal analysis, Investigation, Methodology; Yolande EI O'Donnell, Monica Phimmachanh, Investigation; Alexander D Murphy, Adnan Nagrial, Dariush Daneshvar, Resources, Data curation; Venessa Chin, Resources, Data curation, Supervision, Investigation, Methodology, Project administration; D Neil Watkins, Conceptualization, Supervision, Methodology; Andrew Burgess, Conceptualization, Formal analysis, Supervision, Validation, Investigation, Visualization, Methodology, Project administration; David R Croucher, Conceptualization, Formal analysis, Supervision, Funding acquisition, Investigation, Visualization, Methodology, Project administration

### Author ORCIDs

Andrew Burgess https://orcid.org/0000-0003-4536-9226
David R Croucher https://orcid.org/0000-0003-4965-8674

### Ethics

Human subjects: The analysis of lung adenocarcinoma patient samples and associated clinical data was approved by the St Vincent's Hospital Human Research Ethics Committee (LNR/18/SVH/6). This project was assessed as low/negligible risk and a waiver was therefore granted upon the usual requirement for the consent of the individual to use their health information in a research project. Within the protocol, it was also clearly stated that the results of this study will be published in peer reviewed journals.

Animal experimentation: All experiments were carried out in compliance with the Australian code for the care and use of animals for scientific purposes and in compliance with Garvan Institute of Medical Research/St. Vincent's Hospital Animal Ethics Committee guidelines (ARA_18_17).

Decision letter and Author response
Decision letter https://doi.org/10.7554/eLife.53367.sa1
Author response https://doi.org/10.7554/eLife.53367.sa2

## Additional files

### Supplementary files

• Supplementary file 1. Protein analytes previously implicated in the response to cisplatin.

• Supplementary file 2. Mutation status of the lung adenocarcinoma cell panel.

• Supplementary file 3. Summary of the analytes used for multiplex signalling analysis.

• Supplementary file 4. Characteristics of the patient cohort used to generate immuno-histochemical data.

• Supplementary file 5. Mutation status of second lung adenocarcinoma cell panel.

• Transparent reporting form

### Data availability

All data generated or analysed during this study are included in the manuscript and supporting files.

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
