## [Decision Letter]

**Acceptance summary:**

In this work, the authors generate a model for cisplatin-mediated chemotherapy that more accurately reflects the in vivo pharmacokinetics of this therapeutic. Using this model they identify p70S6K signaling as a key determinant of sensitivity to cisplatin in lung cancer, thus validating the potential to use in vitro models in order to predict and understand therapy resistance.

**Decision letter after peer review:**

Thank you for submitting your article "Analysis of pulsed cisplatin signalling dynamics identifies effectors of resistance in lung adenocarcinoma" for consideration by *eLife*. Your article has been reviewed by three peer reviewers, one of whom is a member of our Board of Reviewing Editors, and the evaluation has been overseen by Maureen Murphy as the Senior Editor. The reviewers have opted to remain anonymous.

The reviewers have discussed the reviews with one another and the Reviewing Editor has drafted this decision to help you prepare a revised submission.

Summary:

In this study, the authors utilized an in vitro model that mimics more physiologically relevant conditions of platinum-based chemotherapy. They achieved this by treating a panel of NSCLC cell lines with a 2 hours pulse of platinum-based chemotherapy and analyzing the effects at 72 hours after the pulse. This is to avoid off-target effects after a prolonged 72 hr period of continuous cisplatin treatment that can create and impair the correct analysis of the mechanisms of sensitivity and resistance to therapy. The authors originally apply a multidimensional analysis of signaling networks generating correlative evidence that can be functionally investigated. Based on these original approaches, the authors find a role of p70S6K in mediating resistance of NSCLC cell to platinum-based chemotherapy, while not confirming some other mechanisms previously identified in cells subjected to a continuous platinum treatment. Moreover, the authors find that in NSCLC patients, the expression of p70S6K correlates with a worse prognosis and worse response to chemotherapy, independently of the mutational status of the *TP53* gene. Using in vitro and mouse xenograft experiments, the authors show that the combination of chemotherapy with inhibitors of p70S6K or si-p70S6K increases sensitivity to platinum. The work is original, well performed and logically structured. However, the reviewers feel that the mechanistic insight is confounded by the reliance on a relatively non-specific small molecule. Improvement of the mechanistic insights through more specific perturbations would improve the rigor and impact of this manuscript. Additionally, the reviewers request that more attention to the flow of the manuscript be paid, and that the authors explain the data presented in a more streamlined and easy to follow manner.

Essential revisions:

1) As pointed out by the authors, dactolisib inhibits the activity of many targets other than P70S6K, including "a number of other PI3K/mTOR substrates." That is quite a list of potential off-target mechanisms. The single experiment in vitro looking at knockdown of P70S6K is encouraging, but it is disappointing that the authors did not validate many of the other effects using this potentially more specific perturbation, instead relying on dactolisib. Further validation with a more specific inhibitor and/or knockdown of P70S6K with a minimum of two short hairpins or siRNAs is recommended to support their many other conclusions, including the cell cycle analysis and also preferably the mouse xenograft experiments.

2) The experiments in Figure 1 comparing the 2 hr vs. 72 hr cisplatin treatment is a bit overstated. It is not surprising that the effects of the two different treatments would be distinct in terms of DNA damage, cell signaling activities, or cell fate. Focusing on the effects of the 2 hr treatment is commendable as being more physiologically relevant and is worthwhile including as a main figure; the 72 hr comparison is better as a supplementary figure.

3) Going all the way to PC3 and PC4 to identify an effect is a little unusual, as those principal components should indicate rather modest contributions. This is particularly concerning given that subsequent validations for the most part rely on a more broad-spectrum inhibitor, which could affect many signaling molecules with higher contributions to PC1 and PC2. Are there significant components of PC1 and PC2 that are affected by dactolisib?

4) The flow of the manuscript is uneven, and the manuscript needs a significant amount of attention: For several figures, the number of figures mentioned in the text does not match the described figures, and the presentation of data is confusing. For example, there is no Figure 1G that the authors keep referring to in the text. Explain better Figure 1E. What is IF ratio? Explain better plots in Figure 2 and Figure 3: what does it mean "component 1, component 2 etc?" in what do these components represent? Also, it is a bit confusing why the response to cisplatin in NSCLC cell lines is independent on the *TP53* status, while the response to p70S6K inhibitors is dependent on it. Can the authors explain this point better? Additional experimental work is required to support such evidence.

Figure 1B and D: It may be better to use a proliferation assay (MTT or EdU) instead of the number of cells. Also, for Figure 4E and F, add statistics. In the subsection “Response to dactolisib is dependent upon *TP53* status”, the link between p21, γ-H2AX and casp3 is not clear. In the third paragraph of the Introduction, the description of previous findings of Fey et al., 2015, is not clear. Clarify the relative sentence. Explain better the choice of molecules belonging to pathways previously implicated in the response to continuous cisplatin exposure, listing them in a table. Explain the function of MCL^-^1 in this context. Along the same lines, explain better the multi-dimensional analysis. It is difficult to interpret.

In the subsection “Continuous versus pulsed cisplatin treatment”, the authors show that many signaling pathways activated upon continuous exposure of cells with platinum, are not activated after the 2 hours pulse at the 72h time point. The authors consequently define the previous observations of other researcher as "artifacts". The authors need to replace the term 'artifact' with something more professional, such as "previous data that should be reconsidered in the light of these new findings". As another example, the reference to Figure 1C at the end of the subsection “Response to cisplatin is not associated with either *TP53* status or drug-efflux”, should be reference to Figure 1E? The axis labels in Figures 2A and D are in odd positions. The axes in Figures 2B and D are unlabeled. Caption for Figure 2D should read, "Visualization of component 3 against component 4…" In the subsection “Validation of model-based observations”, the live cell assay for measuring caspase activity should be better explained in the main text and the caption for Figure 3A. The explanation in the legend for Figure 2—figure supplement 1B is sufficient, it's just confusing because that comes later in the manuscript than the description of the experiment in Figure 3A. The statement "in the absence of p53, we observed that this can be mediated by the related transcription factor p63" is too strongly worded and not fully supported by the findings. A correlation has been shown, not causation. This statement needs to be reworded more cautiously.

5) In Figure 5, the authors show that the p53 null NCI-H358 cells had greater G2 arrest compared to the p53 WT A549 cells following cisplatin pulsing, which they attribute to varying p53 status. To show that this trend is not due to some other factor that differs between these cell lines, the authors should test this after a p53 KO or p53 knockdown with siRNAs or shRNAs in A549 or p53 overexpression in NCI-H358 cells to see if the effect of p53 status is consistent within the same cell line. Also, Figure 7 needs to be toned down especially for the p53 null cells where the authors propose p63 as an effector of cisplatin. For the P70S6K knockdown in Figure 3G, what is the effect on cisplatin sensitive cell lines? This is an important control.

---

## [Author Response]

Essential revisions:1) As pointed out by the authors, dactolisib inhibits the activity of many targets other than P70S6K, including "a number of other PI3K/mTOR substrates." That is quite a list of potential off-target mechanisms. The single experiment in vitro looking at knockdown of P70S6K is encouraging, but it is disappointing that the authors did not validate many of the other effects using this potentially more specific perturbation, instead relying on dactolisib. Further validation with a more specific inhibitor and/or knockdown of P70S6K with a minimum of two short hairpins or siRNAs is recommended to support their many other conclusions, including the cell cycle analysis and also preferably the mouse xenograft experiments.

To further examine the specific role of P70S6K in the response to cisplatin we have now performed the FUCCI imaging analysis in the resistant NCI-H358 cell line with two siRNAs targeting P70S6K (Figure 5—figure supplement 3). In line with the effects observed with dactolisib in this assay, we observed a non-significant trend towards a shortened G2-arrest, and an increase in apoptosis in cells treated with cisplatin following knockdown of P70S6K, an effect that is consistent between each siRNA.

One potential limitation in this assay is the heterogeneity of transient siRNA-mediated knockdown that will be achieved at the single-cell level. While this may have limited our ability to generate significant results using single-cell analysis such as FUCCI, this new data still reinforces two key observations from the original manuscript. Firstly, it further suggests that P70S6K has a specific role in maintaining a protective G2 arrest following cisplatin treatment within platinum resistant lung adenocarcinoma cells. Secondly, it highlights the role of dactolisib mediated P70S6K inhibition in sensitizing resistant cells to this cisplatin treatment.

Because of this result, we also tested the specific P70S6K inhibitor LY2584702 within our cell lines. However, following a one hour incubation with this inhibitor we did not observe any inhibition of P70S6K, as measured by S6 phosphorylation. Using a longer incubation time of 24 hours we observed ~10% inhibition at a concentration of 1 µM, and ~60% inhibition with a concentration of 10 µM. Therefore, due to this poor efficacy we did not proceed with any further experiments using this inhibitor.

Therefore, the addition of the P70S6K siRNA data has resulted in the following changes in the manuscript text:

“To further confirm the specific role of P70S6K in this cell cycle phenotype, this assay was repeated along with siRNA mediated P70S6K knockdown in the resistant NCI-H358 cell line (Figure 5—figure supplement 3). […] While the non-significant decrease in G2-arrest observed upon siRNA-mediated knockdown of P70S6K may have been limited by potential heterogeneity in the level of knockdown achieved, the increase in cell death further suggests that dactolisib mediated inhibition of P70S6K is responsible for the sensitisation of lung adenocarcinoma cells to cisplatin treatment.”

2) The experiments in Figure 1 comparing the 2 hr vs. 72 hr cisplatin treatment is a bit overstated. It is not surprising that the effects of the two different treatments would be distinct in terms of DNA damage, cell signaling activities, or cell fate. Focusing on the effects of the 2 hr treatment is commendable as being more physiologically relevant and is worthwhile including as a main figure; the 72 hr comparison is better as a supplementary figure.

As suggested, we have removed panels A, B and C from Figure 1. These have now become a new Figure 1—figure supplement 1.

3) Going all the way to PC3 and PC4 to identify an effect is a little unusual, as those principal components should indicate rather modest contributions. This is particularly concerning given that subsequent validations for the most part rely on a more broad-spectrum inhibitor, which could affect many signaling molecules with higher contributions to PC1 and PC2. Are there significant components of PC1 and PC2 that are affected by dactolisib?

We agree that it is intriguing that analyzing PC3 and PC4 in this way gave a better distinction between resistant and sensitive cell lines. Within this analysis, PC1 and PC2 were mostly dominated by p53 pathway components (i.e. p53, p-p53, p21, MDM2), which displayed a highly dynamic and variable response between the cell lines, but did not ultimately correlate with the apoptotic response to cisplatin. However, as already demonstrated in Figure 6, dactolisib has a significant effect upon cisplatin mediated expression of p53, p63 and p21. Therefore, while it does not dominate PC1 or PC2, it is likely this ability of P70S6K to finely tune the p53 response that underlies its ability cause resistance to cisplatin in a *TP53* status dependent manner.

To provide clarity on the nuanced role of P70S6K we have made the following alterations to the text describing the proposed mechanism within the Results section:

“While *TP53* mutation status alone is not sufficient to determine sensitivity to cisplatin (Figure 1B), this dependence upon *TP53* mutation status for dactolisib-mediated sensitisation to cisplatin treatment is likely explained by considering the potential role of p21 in maintaining a G2-arrest or promoting an aberrant G2-exit, and thereby preventing DNA damage induced apoptosis occurring during, or shortly after mitosis. […] However, in the *TP53* mutant NCI-H1573 line, which was not sensitised by dactolisib, p21 expression was not significantly elevated following cisplatin treatment, meaning that γH2A.X expression and caspase 3 cleavage were unchanged by the addition of dactolisib (Figure 7).”

4) The flow of the manuscript is uneven, and the manuscript needs a significant amount of attention: For several figures, the number of figures mentioned in the text does not match the described figures, and the presentation of data is confusing.

We apologize for these formatting errors, each has been corrected as indicated below.

For example, there is no Figure 1G that the authors keep referring to in the text. As another example, the reference to Figure 1C at the end of the subsection “Response to cisplatin is not associated with either TP53 status or drug-efflux”, should be reference to Figure 1E?

Our apologies, the referencing of Figure 1 within the original text did contain a number of errors. This is now all correct according to the changes made to the updated Figure 1 and Figure 1—figure supplement 1.

Explain better Figure 1E. What is IF ratio?

The quantification of the imaging of cisplatin-DNA adducts presented in Figure 1C (formerly 1E) was originally labelled as ‘IF ratio’, however this was not clearly explained in the figure legend. We have now altered the labelling of these axes to read ‘Normalised Nuclear Intensity’, and altered the associated figure legend to read:

“(C) Representative images of anti-cisplatin antibody staining in A549 cells following a cisplatin pulse, and quantification of nuclear cisplatin-DNA adducts across the cell line panel (n≥100, mean ± SD). […] All treatment conditions (red) are significantly different from control (blue), p<0.001.”

Explain better plots in Figure 2 and Figure 3: what does it mean "component 1, component 2 etc?" in what do these components represent?

Each of these components represent the principal components which were derived from the Principal Component Analysis (PCA) performed with the data presented in Figure 1D. We have now altered the presentation of these figures, the wording of the figure legend and the descriptions of the components within the text to more clearly represent the identity of each principal component. The updated figure legend now reads:

“Figure 2: Principal component analysis. […] (E) Distribution of the analytes according to their weighting within PC3 and PC4.”

Also, it is a bit confusing why the response to cisplatin in NSCLC cell lines is independent on the TP53 status, while the response to p70S6K inhibitors is dependent on it. Can the authors explain this point better? Additional experimental work is required to support such evidence.

The answer to this reviewer’s comment is similar to the one above describing the role of P70S6K in modulating significant components from PC1 and PC2. Therefore, as described above, to further expand upon the interplay between *TP53* status and P70S6K we have expanded the explanation of the data from Figure 6, which details the proposed mechanism of dactolisib mediated sensitization through the actions of P70S6K on the p53 pathway, which is summarized in Figure 7. This section now reads:

“While *TP53* mutation status alone is not sufficient to determine sensitivity to cisplatin (Figure 1B), this dependence upon *TP53* mutation status for dactolisib-mediated sensitisation to cisplatin treatment is likely explained by considering the potential role of p21 in maintaining a G2-arrest or promoting an aberrant G2-exit, and thereby preventing DNA damage induced apoptosis occurring during, or shortly after mitosis. […] However, in the *TP53* mutant NCI-H1573 line, which was not sensitised by dactolisib, p21 expression was not significantly elevated following cisplatin treatment, meaning that γH2A.X expression and caspase 3 cleavage were unchanged by the addition of dactolisib (Figure 7).”

However, this point is also further elaborated within the following paragraph, which reads:

“In agreement with our finding that elevated P70S6K activity promotes resistance in a *TP53* dependent manner, combination therapy with cisplatin and dactolisib significantly sensitised the high pP70S6K/*TP53* wildtype NCI-H292 cell line, but not the high pP70S6K/*TP53* mutant NCI-H2009 line. Therefore, this data demonstrates that P70S6K promotes an aberrant G2-exit or prolonged G2-arrest following cisplatin treatment, presumably through enhanced translation of p53/p21 in *TP53* wildtype cell lines and p63/p21 in *TP53* null lines (Figure 7). However, in the presence of mutant p53, P70S6K is not able to influence this DNA damage induced signalling axis and is therefore not associated with sensitivity to cisplatin.”

Figure 1B and D: It may be better to use a proliferation assay (MTT or EdU) instead of the number of cells.

For the previous Figure 1B (now Figure 1—figure supplement 1B), we have now included an additional figure (Figure 1—figure supplement 1C), which is an MTS assay demonstrating the difference between the continuous and pulsed cisplatin models. This indirect measure of cell number based upon metabolic activity has indeed given highly similar results to the direct readout of cell number as presented in Figure 1B. However, we disagree that this readout of cell metabolism should be used for the experiment in Figure 1D (now Figure 1B). The purpose of this data was to directly measure the percentage of cells undergoing apoptosis in response to a pulse of cisplatin, this cannot be achieved with an MTS assay.

The addition of this MTS assay has led to the following changes in the manuscript text:

“In order to directly compare the response of lung adenocarcinoma cells to the continuous presence of cisplatin, or a pulse of cisplatin that mimics in vivo pharmacokinetics (2h, 5 μg/mL) (Figure 1—figure supplement 1A), we monitored the growth and apoptosis of the innately resistant A549 lung adenocarcinoma cell line (Marini et al., 2018) by both live cell imaging (Figure 1—figure supplement 1B) and a cell viability assay (Figure 1—figure supplement 1C), under both conditions.”

Also, for Figure 4E and F, add statistics.

The relevant p values have now been included within Figure 4E and F.

In the subsection “Response to dactolisib is dependent upon TP53 status”, the link between p21, γ-H2AX and casp3 is not clear.

As above, we have further expanded upon the proposed mechanism within the Results section describing the data within Figure 6. This now provides more explanation of the mechanistic link between p21 expression and markers of DNA damage and apoptosis:

“While *TP53* mutation status alone is not sufficient to determine sensitivity to cisplatin (Figure 1B), this dependence upon *TP53* mutation status for dactolisib-mediated sensitisation to cisplatin treatment is likely explained by considering the potential role of p21 in maintaining a G2-arrest or promoting an aberrant G2-exit, and thereby preventing DNA damage induced apoptosis occurring during, or shortly after mitosis. […] However, in the *TP53* mutant NCI-H1573 line, which was not sensitised by dactolisib, p21 expression was not significantly elevated following cisplatin treatment, meaning that γH2A.X expression and caspase 3 cleavage were unchanged by the addition of dactolisib (Figure 7).”

In the third paragraph of the Introduction, the description of previous findings of Fey et al., 2015, is not clear. Clarify the relative sentence.

The purpose of this sentence is to highlight that important information about patient-specific drug response can be determined by analyzing the dynamics of drug induced signalling. However, upon re-reading this paragraph in light of the reviewer’s comment, we realise that this sentence is better placed at the beginning of the next paragraph. This following paragraph now reads:

“We have previously shown that predictive computational models of drug-induced apoptotic signalling dynamics can be used as a prognostic indicator of neuroblastoma patient survival (Fey et al., 2015). To move towards a similar concept to platinum resistance in lung adenocarcinoma, we now present an in-depth analysis of the dynamic signalling response to a pulse of platinum chemotherapy, describing the relationship between a number of key signalling nodes, the DNA damage response and platinum sensitivity.”

Explain better the choice of molecules belonging to pathways previously implicated in the response to continuous cisplatin exposure, listing them in a table. Explain the function of MCL^-^1 in this context.

The rationale for inclusion of each of the proteins analysed in Figure 1—figure supplement 1D is now presented within Supplementary file 1. This includes detail on the expected function of MCL^-^1 in the context of cisplatin treatment.

Along the same lines, explain better the multi-dimensional analysis. It is difficult to interpret.

To better explain the purpose and results of the principal component analysis we have added extra detail to the respective paragraph. This text now reads:

“As we had already determined that *TP53* status alone was not sufficient to explain resistance to cisplatin (Figure 1B), we further analysed the whole dataset by performing a principal component analysis (PCA) (Figure 2). This form of dimensionality reduction can be used to identify correlative relationships between variables within a large dataset, and here we have used it to create a visual representation of the association between key signalling nodes and the response to cisplatin across the entire cell line panel.”

In the subsection “Continuous versus pulsed cisplatin treatment”, the authors show that many signaling pathways activated upon continuous exposure of cells with platinum, are not activated after the 2 hours pulse at the 72h time point. The authors consequently define the previous observations of other researcher as "artifacts". The authors need to replace the term 'artifact' with something more professional, such as "previous data that should be reconsidered in the light of these new findings".

This statement has been pulled back to read:

“Taken together, this data demonstrates that previous mechanisms of platinum resistance established using a continuous exposure model should be reconsidered in the light of these new findings regarding the response to physiological levels of drug exposure.”

The axis labels in Figures 2A and D are in odd positions. The axes in Figures 2B and D are unlabeled.

We have now reformatted Figure 2 to make the labelling clearer and more in line with the standardized presentation of PCA plots.

Caption for Figure 2D should read, "Visualization of component 3 against component 4…"

This error has been corrected as suggested.

In the subsection “Validation of model-based observations”, the live cell assay for measuring caspase activity should be better explained in the main text and the caption for Figure 3A. The explanation in the caption for Figure 2—figure supplement 1B is sufficient, it's just confusing because that comes later in the manuscript than the description of the experiment in Figure 3A.

We have altered the figure caption to read:

“(A) An overlay of real-time imaging of apoptosis following a cisplatin pulse, performed on the Incucyte platform using a fluorescent caspase substrate (1 µM), on top of the PCA plot of component 3 against component 4.”

We have also altered the manuscript text to read:

“The association between this signalling state and increased sensitivity to cisplatin can also be clearly observed by overlaying an orthogonal readout of the apoptotic response onto this PCA plot (Figure 3A). This real-time apoptosis data was generated using a fluorescent caspase substrate as an indicator of cell death across the cell line panel for 72 h following the cisplatin pulse treatment.”

The statement "in the absence of p53, we observed that this can be mediated by the related transcription factor p63" is too strongly worded and not fully supported by the findings. A correlation has been shown, not causation. This statement needs to be reworded more cautiously.

We have pulled this statement back by changing the text to:

“However, in the absence of p53, we observed that this may potentially be mediated by the related transcription factor p63.”

5) In Figure 5, the authors show that the p53 null NCI-H358 cells had greater G2 arrest compared to the p53 WT A549 cells following cisplatin pulsing, which they attribute to varying p53 status. To show that this trend is not due to some other factor that differs between these cell lines, the authors should test this after a p53 KO or p53 knockdown with siRNAs or shRNAs in A549 or p53 overexpression in NCI-H358 cells to see if the effect of p53 status is consistent within the same cell line.

As suggested, we have performed the FUCCI cell cycle analysis following p53 knockdown in the A549 cell line (Figure 5—figure supplement 2). Importantly, as previously demonstrated with the A549 wildtype and NCI-H358 null lines, there was a significant increase in G2 arrest and a significant decrease in G2 exit within the p53 knockdown cells, when compared to the negative control siRNA treated cells. Therefore, this data corroborates the initial analysis and further demonstrates that differences in the length of G2 arrest or occurrence of an aberrant G2 exit observed upon cisplatin treatment can be attributed to the varying *TP53* mutation status across these cell lines.

The addition of this data has resulted in the following changes in the manuscript text:

“Therefore, to confirm the specific role of p53 in this context, this assay was repeated with an siRNA-mediated knockdown of p53 in A549 cells (Figure 5—figure supplement 2). In line with this hypothesis, this orthogonal approach revealed a significant increase in G2-arrest in the p53 knockdown cells, along with a decrease in occurrence of G2-exit, effectively pheno-copying the observed difference between the A549 wildtype and NCI-H358 p53 null cell lines (Figure 5C).”

Also, Figure 7 needs to be toned down especially for the p53 null cells where the authors propose p63 as an effector of cisplatin.

In line with the additional comment above, we have now altered this figure to demonstrate that p63 may potentially be able to mediate p21 expression in the p53 null cells.

For the P70S6K knockdown in Figure 3G, what is the effect on cisplatin sensitive cell lines? This is an important control.

To address this comment we performed P70S6K knockdown in the cisplatin sensitive NCI-H1299 cell line, in the same manner as performed in Figure 3G. In line with their existing sensitivity to cisplatin treatment, and that dactolisib did not result in any further sensitization, P70S6K knockdown did not further sensitise the NCI-H1299 cells.

This data is now presented in Figure 3—figure supplement 1, and resulted in the following alterations to the manuscript text:

“Under these conditions, cisplatin treatment resulted in significantly higher caspase 3 cleavage and γH2A.X expression in the P70S6K knockdown cells (Figure 3G), whilst P70S6K knockdown did not further sensitise the already cisplatin sensitive NCI-H1299 cell line (Figure 3—figure supplement 1).”